# Rigidity-Aware Geometric Pretraining for Protein Design and Conformational Ensembles

**Zhanghan Ni**[1*]    **Yanjing Li**[2*]    **Zeju Qiu**[3*]    **Bernhard Schölkopf**[3]    **Hongyu Guo**[4,5]
**Weiyang Liu**[3,6]    **Shengchao Liu**[6]

[1]University of Illinois Urbana-Champaign    [2]University of Washington
[3]MPI for Intelligent Systems, Tübingen    [4]National Research Council of Canada
[5]University of Ottawa    [6]The Chinese University of Hong Kong    [*]Equal contribution

## Abstract

Generative models have recently advanced *de novo* protein design by learning the statistical regularities of natural structures. However, current approaches face three key limitations: (1) Existing methods cannot jointly learn protein geometry and design tasks, where pretraining can be a solution; (2) Current pretraining methods mostly rely on local, non-rigid atomic representations for property prediction downstream tasks, limiting global geometric understanding for protein generation tasks; and (3) Existing approaches have yet to effectively model the rich dynamic and conformational information of protein structures. To overcome these issues, we introduce **RigidSSL** (*Rigidity-Aware Self-Supervised Learning*), a geometric pretraining framework that front-loads geometry learning prior to generative finetuning. Phase I (RigidSSL-Perturb) learns geometric priors from 432K structures from the AlphaFold Protein Structure Database with simulated perturbations. Phase II (RigidSSL-MD) refines these representations on 1.3K molecular dynamics trajectories to capture physically realistic transitions. Underpinning both phases is a bi-directional, rigidity-aware flow matching objective that jointly optimizes translational and rotational dynamics to maximize mutual information between conformations. Empirically, RigidSSL variants improve designability by up to 43% while enhancing novelty and diversity in unconditional generation. Furthermore, RigidSSL-Perturb improves the success rate by 5.8% in zero-shot motif scaffolding and RigidSSL-MD captures more biophysically realistic conformational ensembles in G protein-coupled receptor modeling. The code is available on this repository.

## 1 Introduction

Proteins are large and complex biomolecules whose three-dimensional structures determine their diverse biological functions, enabling precise molecular activities such as substrate specificity, binding affinity, and catalytic activity (LaPelusa & Kaushik, 2025). The ability to design proteins with desired properties has the potential to revolutionize multiple disciplines. Examples include novel therapeutics and vaccines in medicine and sustainable biomaterials in materials science (Listov et al., 2024; Miserez et al., 2023). In recent years, rapid advances in deep generative modeling have opened new directions for protein science (Huguet et al., 2024; Yim et al., 2023). More specifically, generative modeling has been widely used for protein-related biology tasks, including *de novo* protein design (Bose et al., 2023; Watson et al., 2023), motif scaffolding (Wang et al., 2022a), and ensemble generation (Jing et al., 2024), showing the potential of geometric learning to model structure and dynamics at scale.

*Challenge 1: Learning geometry and generation simultaneously brings in challenges in model generalization.* Despite significant progress in protein generative modeling, existing end-to-end frameworks often require the model to jointly learn the fundamental geometry of proteins and the complex mechanism of structure generation within a single objective (Huguet et al., 2024; Yim et al., 2023). This tight coupling can lead to inefficient optimization and limit generalization to novel or out-of-distribution design tasks. To alleviate this issue, existing approaches typically follow three directions. One line of work repurposes large structure prediction models to inherit their geometric priors (Watson et al., 2023), but this strategy incurs substantial computational overhead and limits

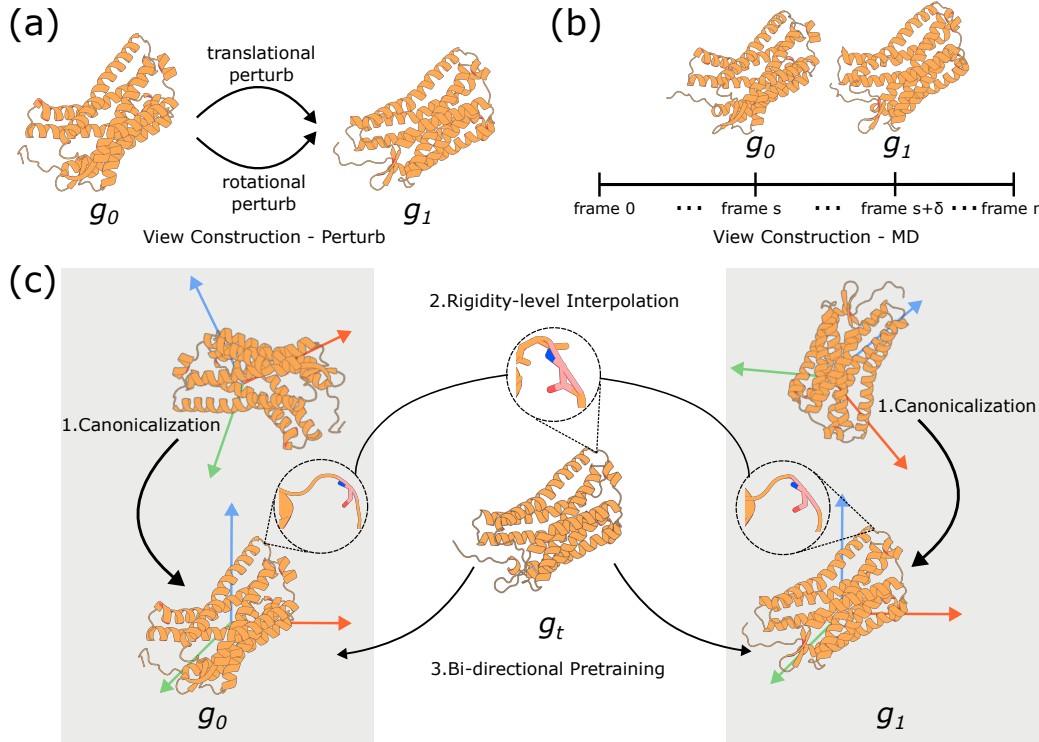

Figure 1: Overview of RigidSSL. (a) View construction in RigidSSL-Perturb: translational noise in $\mathbb{R}^3$ and rotational noise in $SO(3)$ are applied to generate perturbations in the rigid body motion group $SE(3)$. (b) View construction in RigidSSL-MD: perturbed states are obtained by sampling conformational frames from MD trajectories. (c) Rigidity-based pretraining in RigidSSL: proteins are canonicalized into a reference frame, intermediate states are constructed via interpolation of translations and rotations for each rigid residue frame, and bi-directional flow matching is applied for pretraining. Details can be found in Section 3.

architectural flexibility. Another line of work designs sophisticated architectures to compensate for the lack of priors (Wang et al., 2024), resulting in complex and specialized models to achieve competitive performance. Motivated by the success of pretraining (Jiao et al., 2022; Liu et al., 2021; 2022a; 2023b; Zaidi et al., 2022), a more scalable alternative is to front-load geometric understanding prior to downstream generation by first pretraining models to acquire a general understanding of protein geometry. Such pretrained representations can then serve as a foundation for diverse generative tasks. However, the effectiveness of this paradigm critically depends on how protein geometry is represented, which determines what structural priors can be learned and transferred.

*Challenge 2: Pretraining requires global and efficient representations.* Realizing an effective pretraining paradigm places stringent requirements on the underlying representations, which must support scalable computation while faithfully capturing global geometry. However, most geometric pretraining methods rely on local, non-rigid atomic or fragment-level representations (Chen et al., 2023; Guo et al., 2022; Hermosilla & Ropinski, 2022; Zhang et al., 2022) that primarily capture short-range geometric patterns. While this locality is often sufficient for property prediction, it can under-represent global folding geometry, limiting structural abstraction and thus the transferability of learned representations to generative design tasks (Bouatta et al., 2021; Skolnick et al., 2021). Achieving representations that are both global and efficient is nontrivial, in part because all-atom modeling is computationally expensive, which limits scalability and learning efficiency on large geometric datasets.

*Challenge 3: Pretraining requires diverse and multi-scale data.* In addition to the choice of protein backbone representation, the availability and quality of the dataset present another critical bottleneck for effective structure-based pretraining. Large-scale structure databases such as the AlphaFold Protein Structure Database (AFDB) (Varadi et al., 2022) and Protein Data Bank (PDB) (Berman et al., 2000) together provide hundreds of millions of protein structures encoding rich geometric regularities

of natural folds, which is an invaluable resource for geometric pretraining. However, these datasets are dominated by static structures and largely overlook the intrinsic conformational flexibility of proteins. As a result, models pretrained solely on such static snapshots primarily learn the geometry of static states, failing to capture near-native fluctuations or transitions among metastable conformations. This limitation highlights a fundamental challenge for structure-based pretraining: the requirement for diverse, multi-scale data, including conformational ensembles derived from molecular dynamics (MD) simulations, that span both rigid geometric motifs and dynamic variability to support generative modeling with improved conformational diversity.

**Our contributions.** To address these challenges, we propose **RigidSSL** (*Rigidity-Aware Self-Supervised Learning*), a two-stage geometric pretraining framework for protein structure generation. RigidSSL explicitly follows the backbone modeling via residue-level translations and rotations in $\mathrm{SE}(3)$, which substantially reduces degrees of freedom and allows the model to learn geometric priors under physical constraints. Building on such representation, RigidSSL employs a **two-phase pretraining strategy** that sequentially integrates multi-scale structural information from static and dynamic data. **Phase I (RigidSSL-Perturb)** pretrains on 432K static AFDB structures with simulated perturbations in $\mathrm{SE}(3)$ to emulate broad but coarse conformational variation (Figure 1(a)); **Phase II (RigidSSL-MD)** pretrains on 1.3K MD trajectories to provide physically realistic transitions, refining the representation toward true dynamical flexibility (Figure 1(b)). In both phases, operating under a canonicalized reference system, RigidSSL adopts a bi-directional, rigidity-aware flow matching objective to jointly optimize translational and rotational dynamics, serving as a surrogate for maximizing mutual information between paired conformations (Figure 1(c)).

To verify the effectiveness of RigidSSL, we consider two types of downstream tasks: (1) protein design, including protein unconditional generation and motif scaffolding, and (2) conformational ensemble generation. In unconditional generation, RigidSSL variants improve designability by up to 43% while enhancing novelty and diversity. Notably, RigidSSL-Perturb effectively generalizes to long protein chains of 700–800 residues, yielding the best stereochemical quality as reflected by the lowest Clashscore and MolProbity score. In motif scaffolding, RigidSSL-Perturb improves the average success rate by 5.8%. In ensemble generation, RigidSSL variants learn to represent a wider and more physically realistic conformational landscape of G protein-coupled receptors (GPCRs), collectively achieving the best performance on 7 out of 9 metrics. Overall, RigidSSL substantially improves physical plausibility and generative diversity in downstream design tasks.

**Related Work.** We briefly review the most related works here and include a more detailed discussion in Appendix A. Geometric representation learning has been extensively explored for both small molecules (Coors et al., 2018; Gasteiger et al., 2020; Schütt et al., 2018) and proteins (Fan et al., 2022; Jing et al., 2020; Wang et al., 2022b), with methods categorized into $\mathrm{SE}(3)$-invariant and $\mathrm{SE}(3)$-equivariant models (Liu et al., 2023a). Existing pretraining methods for small molecules optimize mutual information between or within modalities (Jiao et al., 2022; Liu et al., 2021; 2022a; 2023b; Zaidi et al., 2022). Protein-specific pretraining has evolved through diffusion-based methods (Guo et al., 2022), contrastive learning on substructures via GearNet (Zhang et al., 2022) and ProteinContrast (Hermosilla & Ropinski, 2022), and MSA-based strategies like MSAGPT (Chen et al., 2024). Finally, joint protein-ligand pretraining approaches, including CoSP (Gao et al., 2022) and MBP (Yan et al., 2023), model binding interactions directly.

## 2 Preliminaries

**Protein Structure Representation.** A protein is composed of a sequence of residues, each containing three backbone atoms in the order $N$, $C_\alpha$, and $C$. Accordingly, the protein backbone with $L$ residues can be represented as $L \times [N, C_\alpha, C]$. Although there is rotational freedom around the $N$-$C_\alpha$ ($\phi$) and $C_\alpha$-$C$ ($\psi$) bonds, following the common simplification used in AlphaFold2 (Jumper et al., 2021), we treat each residue as a rigid body. In this formulation, bond lengths and bond angles within the backbone are fixed to idealized values, and only torsional rotations ($\phi, \psi, \omega$) are allowed to vary. Because each residue is modeled as a rigid body, its configuration in 3D space can be described entirely by its position and orientation. Thus, a protein chain can be modeled as a sequence of rigid residues, where each residue is parameterized by a translation and a rotation. Formally, we represent a protein structure as a sequence of rigid transformations $g_{\mathrm{raw}} = \{T_{\mathrm{raw},i}\}_{i=1}^{L} = \{(\vec{t}_{\mathrm{raw},i}, r_{\mathrm{raw},i})\}_{i=1}^{L}$, where the subscript "raw" indicates the raw and uncanonicalized coordinates from the database,

and each residue $i$ is described by a translation vector $\vec{t}_i \in \mathbb{R}^3$ and a rotation matrix $r_i \in \mathrm{SO}(3)$, specifying its position and orientation in 3D space. Later, we will adopt a canonicalization process to align each protein structure to its invariant pose, *i.e.*, $g = \{T_i\}_{i=1}^L = \{(\vec{t}_i, r_i)\}_{i=1}^L$. We define $x_i := \vec{t}_i$ as the 3D coordinates of the $C_\alpha$ atom at residue $i$, so $\{x_i\}_{i=1}^L$ gives the $C_\alpha$ trace of the backbone. In what follows, we will use $\vec{t}_i$ and $x_i$ interchangeably according to different contexts. Furthermore, each residue is also associated with an amino acid identity $A_i \in \{1, \ldots, 21\}$, representing the 20 standard amino acids plus one unidentified type. More details can be found in Appendix C.1.

**Structure Encoder.** Invariant Point Attention (IPA), first introduced in AlphaFold2 (Jumper et al., 2021), is a geometric attention mechanism designed to encode protein backbone representations. It is provably invariant under rigid Euclidean transformations due to the fact that the inner product of two rigidity bases stays constant under SE(3) transformations. We adopt IPA as the base model in both pretraining and downstream tasks. More details can be found in Appendix E.

**Flow Matching.** Flow Matching (FM) has been an expressive generative model (Albergo & Vanden-Eijnden, 2022; Lipman et al., 2022; Liu et al., 2022b). It directly minimizes the discrepancy between a learnable vector field $v_t(x)$ and a target velocity field $u_t(x)$ that transports samples along a probability path from a simple prior distribution at $t = 0$ to a target distribution at $t = 1$. The training objective is

$$\mathcal{L}_{\mathrm{FM}}(\theta) = \mathbb{E}_{t \sim \mathcal{U}[0,1],\, x \sim p_t(x)} \left[ \|v_t(x) - u_t(x)\|^2 \right], \tag{1}$$

where $v_t(x)$ is the model output and $u_t(x)$ is the ideal (but generally unknown) velocity field that induces the time-dependent distribution $p_t(x)$. However, this objective is often intractable because computing $u_t(x)$ requires knowledge of the exact marginal distributions and their dynamics.

To address this issue, Conditional Flow Matching (CFM) (Lipman et al., 2022) introduces conditioning on a known target sample $x_1$ to define a tractable conditional probability path from $x_0$ to $x_1$. The objective then becomes:

$$\mathcal{L}_{\mathrm{CFM}}(\theta) = \mathbb{E}_{t \sim \mathcal{U}[0,1],\, x_1 \sim q(x_1),\, x \sim p_t(x|x_1)} \left[ \|v_t(x) - u_t(x; x_1)\|^2 \right], \tag{2}$$

where $q(x_1)$ denotes the marginal distribution of targets and $p_t(x \mid x_1)$ defines the interpolated distribution between $x_0$ and $x_1$ at time $t$. In this setting, $u_t(x; x_1)$ is a known velocity field (*e.g.*, from linear or spherical interpolation), making the training objective tractable. This formulation enables an efficient training of conditional generative models by leveraging path-based supervision.

## 3 METHOD: RIGIDSSL

RigidSSL learns transferable geometric representations of proteins by integrating large-scale static structures with dynamic conformational data. While databases such as the PDB and AFDB provide high-quality but static snapshots, native proteins continually fluctuate around equilibrium states (Miller & Phillips, 2021; Nam & Wolf-Watz, 2023). To capture both equilibrium geometry and near-native dynamics, RigidSSL adopts a two-phase rigidity-based pretraining strategy (Figure 1), comprising three components: (1) frame canonicalization; (2) view construction; and (3) an objective function based on rigidity-guided flow matching. Two phases in pretraining contain: Phase I (RigidSSL-Perturb) distills large-scale geometric regularities from 432K AFDB structures via rigid body perturbations in SE(3). Phase II (RigidSSL-MD) refines these representations using 1.3K MD trajectories that capture physically realistic motions. Across both phases, proteins are canonicalized into inertial reference frames, and the model is optimized with a bi-directional, rigidity-aware flow matching objective that jointly optimizes translational and rotational dynamics to maximize mutual information between conformations.

### 3.1 REFERENCE FRAME CANONICALIZATION

Protein structures in databases exist under arbitrary coordinate systems. To establish a canonical reference system, we align each protein structure to its reference inertial frame, a convention widely adopted in the community (Guo et al., 2025; Li et al., 2025). Recall from Section 2 that during modeling, we treat each protein structure $g$ as a sequence of $L$ rigid residues, where each residue is represented with a translation and rotation transformation, *i.e.*, $g_{\mathrm{raw}} = \{T_{\mathrm{raw},i}\}_{i=1}^L = \{(\vec{t}_{\mathrm{raw},i}, r_{\mathrm{raw},i})\}_{i=1}^L$. Recall that the subscript "raw" indicates the raw and uncanonicalized coordinates from the database. Based on this, the global alignment proceeds in two steps.

**Translational Alignment.** We align the protein to its center of mass, $\bar{x} = \frac{1}{L}\sum_{i=1}^{L} x_{\text{raw},i}$, which defines the origin of the inertial frame. Each residue's coordinate is shifted as $x_i = x_{\text{raw},i} - \bar{x}$, and the translation vector becomes $\vec{t_i} = \vec{t}_{\text{raw},i} - \bar{x}$.

**Rotational Alignment.** We align the protein to its principal axes, defined by the axes of the inertial frame. More concretely, we compute the inertia tensor $\hat{\mathbf{I}} = \sum_{i=1}^{L} \left( \|x_i\|^2 \mathbf{I}_3 - x_i x_i^\top \right)$, where $\mathbf{I}_3$ is the $3 \times 3$ identity matrix. We then perform the eigendecomposition $\hat{\mathbf{I}} = \mathbf{V}\mathbf{\Lambda}\mathbf{V}^\top$, where the columns of $\mathbf{V}$ are the eigenvectors forming the principal axes of inertia. Additional geometric constraints are applied to obtain a deterministic $\mathbf{V} \in \text{SO}(3)$ (see Appendix C.2). Then, the aligned rotation matrix is computed as $r_i = r_{\text{raw},i} \cdot \mathbf{V}$.

The canonicalized protein structure is represented as $g = \{T_i\}_{i=1}^{L} = \{(\vec{t_i}, r_i)\}_{i=1}^{L}$. Frame alignment serves as a canonicalization step before RigidSSL-Perturb and RigidSSL-MD, as will be introduced next. By expressing all protein structures in a consistent reference frame, we ensure that the rotational and translational interpolation paths in SE(3) also reside in a consistent reference system, improving the generalizability of RigidSSL.

### 3.2 VIEW CONSTRUCTIONS IN A TWO-PHASE PRETRAINING FRAMEWORK

#### 3.2.1 PHASE I: RIGIDSSL-PERTURB

RigidSSL-Perturb adopts the massive AFDB (Varadi et al., 2022) with 432K unique protein structures. In each training iteration, RigidSSL-Perturb applies a simulated perturbation to each AFDB structure $g^0$ to generate a second view $g^1$, as shown in Figure 1(a). Concretely, for $g^0$ parameterized as a collection of *aligned* rigid bodies $\{T_i\}_{i=1}^{L} = \{(\vec{t_i}, r_i)\}_{i=1}^{L}$, RigidSSL-Perturb independently constructs a perturbed view for each rigid body $T_i = (\vec{t_i}, r_i)$ to obtain $g^1$. This residue-wise perturbation handles translation $\vec{t_i}$ and rotation $r_i$ separately.

**Translation Perturbation.** $\vec{t_i}^{0}$ is perturbed by adding a Gaussian noise in the Euclidean space:

$$\vec{t_i}^{1} = \vec{t_i}^{0} + \sigma \cdot z, \quad z \sim \mathcal{N}(0, \mathbf{I}_3). \tag{3}$$

**Rotation Perturbation.** To generate physically plausible structural variations in protein backbones, we employ the isotropic Gaussian distribution on the special orthogonal group $\text{SO}(3)$ ($\mathcal{IG}_{\text{SO}(3)}$). This choice is guided by three key considerations: (1) protein dynamics fundamentally arise from thermal Brownian motion, which $\mathcal{IG}_{\text{SO}(3)}$ naturally models in the rotational domain (Yanagida et al., 2007), (2) rotations form a non-Euclidean manifold, requiring manifold-aware sampling to maintain geometric validity, and (3) since proteins explore their entire conformational landscape through continuous rotational changes, we need smooth, continuous sampling across all rotational states without singularities. Methodologically, $\mathcal{IG}_{\text{SO}(3)}$ is parametrized by a mean rotation $\mu \in \text{SO}(3)$ and a concentration parameter $\epsilon \in \mathbb{R}_+$. The density function of this distribution is expressed as:

$$p(r; \mu, \epsilon^2) = \frac{1}{Z(\epsilon)} \exp\left(\frac{1}{\epsilon^2}\text{trace}(\mu^T r)\right), \tag{4}$$

where $Z(\epsilon)$ is a normalization constant. To sample a rotation $r \sim \mathcal{IG}_{\text{SO}(3)}(\mu, \epsilon^2)$, we follow the axis-angle parameterization and sampling method of Leach et al. (2022) (see Appendices D.2 and D.3). Then, the rotational rigid perturbation is obtained via:

$$r_i^{1} = r_i^{0} \cdot r, \quad r \sim \mathcal{IG}_{\text{SO}(3)}(\mathbf{I}, \epsilon^2). \tag{5}$$

Note that perturbations are applied to canonicalized proteins to ensure consistent geometric effects, as detailed in Appendix D.4.

**Summary.** Building upon Equations (3) and (5), we define a perturbation function $Perturb_{\sigma,\epsilon} : SE(3) \to SE(3)$ that applies both translational and rotational noise to individual rigid body frames:

$$T_i^{1} = Perturb_{\sigma,\epsilon}(T_i^{0}) = (\vec{t_i}^{0} + \sigma \cdot z, r_i^{0} \cdot r), \quad \text{where } z \sim \mathcal{N}(0, \mathbf{I}_3), \ r \sim \mathcal{IG}_{\text{SO}(3)}(\mathbf{I}, \epsilon^2). \tag{6}$$

The perturbed view $g^1$ is then constructed as the set of transformed frames $\{T_i^1\}_{i=1}^{L}$.

### 3.2.2 PHASE II: RIGIDSSL-MD

As for phase II, RigidSSL-MD aims to learn about more realistic protein structural dynamics. It is trained on the ATLAS dataset (Vander Meersche et al., 2024), which contains 1.3K MD trajectories. MD simulations sample the energy landscape of proteins by numerically integrating Newton's equations of motion, producing trajectories that reflect structural fluctuations governed by physical force fields. To capture such fluctuations, we form two views $g^0$ and $g^1$ from time-separated snapshots within the same trajectory.

In a trajectory, the raw snapshot at time $s$ can be denoted as $F_{\text{raw},s} = \{(\vec{t}^s_{\text{raw},i}, r^s_{\text{raw},i})\}_{i=1}^L$. We extract pairs $(F_{\text{raw},s}, F_{\text{raw},s+\delta})$ with a fixed interval $\delta$. The choice of $\delta$ controls the scale of conformational variation: smaller values capture only minor thermal vibrations, whereas larger values may reflect substantial conformational rearrangements. In this work, we set $\delta = 2$ ns to yield views that represent conformational fluctuations. Canonicalizing the raw snapshots gives $F^s, F^{s+\delta}$, yielding $g^0 := F^s = \{(\vec{t}^s_i, r^s_i)\}_{i=1}^L$ and $g^1 := F^{s+\delta} = \{(\vec{t}^{s+\delta}_i, r^{s+\delta}_i)\}_{i=1}^L$.

### 3.3 RIGID FLOW MATCHING FOR MULTI-VIEW PRETRAINING: RIGIDSSL

In Sections 3.2.1 and 3.2.2, we introduce two sequential phases, each constructing a set of paired views $\{(g^0, g^1)\}$. Our goal is to learn a generalizable latent space by maximizing the mutual information (MI) between the constructed views, *i.e.*, $\mathcal{L} = \text{MI}(g^0, g^1)$. It encourages the model to capture the underlying geometric and structural patterns that are preserved across different views.

To address this, we follow Liu et al. (2021) and adopt a surrogate objective based on conditional likelihoods:

$$\mathcal{L} = \log p(g^0 \mid g^1) + \log p(g^1 \mid g^0), \tag{7}$$

where $g^0 = \{(\vec{t}_i^{\,0}, r_i^{\,0})\}_{i=1}^L$ and $g^1 = \{(\vec{t}_i^{\,1}, r_i^{\,1})\}_{i=1}^L$ denote two rigid body views of a protein, with translations $\vec{t}$ and rotations $r$. For later calculation, we convert the rotation matrix $r$ into a quaternion $q$ for interpolation between views. Thus, the objective becomes

$$\mathcal{L} = \log p(\{\vec{t}_i^{\,0}, q_i^{\,0}\} \mid \{\vec{t}_i^{\,1}, q_i^{\,1}\}) + \log p(\{\vec{t}_i^{\,1}, q_i^{\,1}\} \mid \{\vec{t}_i^{\,0}, q_i^{\,0}\}). \tag{8}$$

To optimize Equation (8), we adopt the Flow Matching framework (Albergo & Vanden-Eijnden, 2022; Lipman et al., 2022; Liu et al., 2022b). Unlike generic flows, our model respects the inductive bias that each residue behaves as a rigid body during interpolation. The objective is to learn a velocity field that drives the system from $g^0$ to $g^1$ (and vice versa), via an intermediate state at time $\tau \in [0, 1]$. We decompose this into translation and rotation components, as described below.

**LERP for Translation in** $\mathbb{R}^3$. We interpolate the translations between residues using linear interpolation (LERP):

$$\vec{t}^{\,\tau} = \text{LERP}(\vec{t}^{\,0}, \vec{t}^{\,1}, \tau) = \tau\vec{t}^{\,1} + (1 - \tau)\vec{t}^{\,0}, \tag{9}$$

where $\tau \in [0, 1]$ is the interpolation parameter.

**SLERP for Rotation in** $\text{SO}(3)$. For rotations, we interpolate quaternions using spherical linear interpolation (SLERP):

$$q^\tau = \text{SLERP}(q^0, q^1, \tau) = \frac{\sin((1 - \tau)\phi)q^0 + \sin(\tau\phi)q^1}{\sin(\phi)}, \tag{10}$$

where $\tau \in [0, 1]$ is the interpolation parameter and $\phi$ is the angle between $q^0$ and $q^1$.

Recall that we are using IPA as our backbone model, $v_\theta$. We want $v_\theta$ to learn the true flow of both translation and rotation (quaternion) through flow matching, *i.e.*, $[\mathbf{u}_{\theta,\mathbb{R}^3}, \mathbf{u}_{\theta,\text{SO}(3)}] = v_\theta(\vec{t}^{\,\tau}, q^\tau, \tau)$. Thus, the objective function for one direction $g^0 \rightarrow g^1$ is:

$$\mathcal{L}^{g^0 \rightarrow g^1} = \mathcal{L}_{\mathbb{R}^3} + \mathcal{L}_{\text{SO}(3)} = \left\| \vec{t}^{\,1} - \vec{t}^{\,0} - \mathbf{u}_{\theta,\mathbb{R}^3} \right\|^2 + \left\| \frac{d}{d\tau}\text{SLERP}(q^0, q^1, \tau) - \mathbf{u}_{\theta,\text{SO}(3)} \right\|^2. \tag{11}$$

**Final Objective.** We apply the same formulation in the reverse direction, yielding the final loss:

$$\mathcal{L} = \mathcal{L}^{g^0 \rightarrow g^1} + \mathcal{L}^{g^1 \rightarrow g^0}. \tag{12}$$

The final objective is employed sequentially over the two phases described in Sections 3.2.1 and 3.2.2, which we refer to as RigidSSL-Perturb and RigidSSL-MD, respectively.

Table 1: Comparison of Designability (fraction with scRMSD $\leq 2.0$ Å), Novelty (max. TM-score to PDB), and Diversity (avg. pairwise TM-score and MaxCluster diversity). Reported with standard errors. Best mean values are in **bold** and second-best are underlined.

| Model | Pretraining | Designability | Novelty | Diversity | |
|---|---|---|---|---|---|
| | | Fraction ($\uparrow$) | avg. max TM ($\downarrow$) | pairwise TM ($\downarrow$) | MaxCluster ($\uparrow$) |
| FrameDiff | None | $0.775 \pm 0.060$ | $0.555 \pm 0.079$ | $0.565 \pm 0.016$ | 0.033 |
| FrameDiff | GeoSSL-EBM-NCE | $0.725 \pm 0.048$ | $0.615 \pm 0.054$ | $0.597 \pm 0.010$ | 0.033 |
| FrameDiff | GeoSSL-InfoNCE | $0.650 \pm 0.058$ | $0.613 \pm 0.068$ | $0.568 \pm 0.013$ | 0.033 |
| FrameDiff | GeoSSL-RR | $0.700 \pm 0.065$ | $0.579 \pm 0.073$ | $0.604 \pm 0.015$ | 0.089 |
| FrameDiff | RigidSSL-Perturb (ours) | $\mathbf{0.875 \pm 0.050}$ | $\mathbf{0.494 \pm 0.066}$ | $0.534 \pm 0.011$ | 0.033 |
| FrameDiff | RigidSSL-MD (ours) | $0.700 \pm 0.062$ | $0.657 \pm 0.084$ | $\mathbf{0.471 \pm 0.009}$ | **0.156** |
| FoldFlow-2 | None | $0.329 \pm 0.013$ | $0.810 \pm 0.004$ | $0.620 \pm 0.002$ | 0.183 |
| FoldFlow-2 | GeoSSL-EBM-NCE | $0.424 \pm 0.014$ | $0.790 \pm 0.009$ | $0.626 \pm 0.002$ | 0.225 |
| FoldFlow-2 | GeoSSL-InfoNCE | $0.333 \pm 0.012$ | $0.786 \pm 0.005$ | $0.631 \pm 0.001$ | 0.052 |
| FoldFlow-2 | GeoSSL-RR | $0.344 \pm 0.012$ | $0.787 \pm 0.005$ | $\mathbf{0.601 \pm 0.003}$ | 0.137 |
| FoldFlow-2 | RigidSSL-Perturb (ours) | $\mathbf{0.758 \pm 0.016}$ | $\mathbf{0.770 \pm 0.003}$ | $0.650 \pm 0.001$ | 0.252 |
| FoldFlow-2 | RigidSSL-MD (ours) | $0.584 \pm 0.018$ | $0.782 \pm 0.004$ | $0.613 \pm 0.002$ | **0.318** |

## 4 EXPERIMENTS AND RESULTS

We pretrain the IPA module under the RigidSSL framework, following Appendix F. We evaluate on diverse protein structure generation tasks against four baselines: a randomly initialized model and three coordinate-aware geometric self-supervised learning methods that maximize mutual information. Specifically, GeoSSL-InfoNCE and GeoSSL-EBM-NCE employ contrastive objectives that align positive pairs while repelling negatives (Liu et al., 2021; Oord et al., 2019), whereas GeoSSL-RR adopts a generative objective, leveraging intra-protein supervision by reconstructing one view from its counterpart in the representation space (Liu et al., 2021). A comparison of pretraining and downstream training configurations can be found in Appendix F.1.

### 4.1 DOWNSTREAM: UNCONDITIONAL PROTEIN STRUCTURE GENERATION

We evaluate two representative generative models for protein backbones, FrameDiff and FoldFlow-2, both built on IPA and rigid body representations. FrameDiff (Yim et al., 2023) generates novel backbones by defining a diffusion process directly on the manifold of residue-wise SE(3) frames. Built on IPA to update embeddings, it learns an SE(3)-equivariant score function that drives the reverse diffusion dynamics. FoldFlow-2 (Huguet et al., 2024) is a sequence-augmented, SE(3)-equivariant flow-matching model. Its architecture consists of two IPA blocks, with the first encoding protein structures into latent representations and the second decoding multimodal inputs into $SE(3)_0^L$ vector fields used for backbone generation. We pretrain the IPA module in FrameDiff and FoldFlow-2 with RigidSSL, and subsequently finetune them for monomer backbone generation following their original training objectives (see Appendix F). We then assess the generated samples in terms of designability, diversity, novelty, Fréchet Protein Structure Distance (FPSD), Fold Jensen–Shannon Divergence (fJSD), and Fold Score (fS). See Appendix F.4.2 for metric details.

As shown in Table 1, compared with no pretraining or with GeoSSL-EBM-NCE, GeoSSL-InfoNCE, and GeoSSL-RR, FrameDiff pretrained with RigidSSL-Perturb achieves substantially higher performance in designability, novelty, and diversity. Specifically, relative to the unpretrained FrameDiff, RigidSSL-Perturb improves mean designability by $10\%$, mean novelty by $6.1\%$, and mean diversity by $3.1\%$ (as measured by pairwise TM). However, while RigidSSL-MD further improves mean diversity by $9.4\%$ relative to the unpretrained model, it leads to decreased performance in designability and novelty. We provide further discussion of this phenomenon in Section 5. Similarly, for FoldFlow-2, RigidSSL-Perturb improves mean designability by $42.9\%$, mean novelty by $4\%$, and mean diversity by $6.9\%$ (as measured by MaxCluster) when compared with the unpretrained model. We hypothesize that the increased diversity across FrameDiff and FoldFlow-2 by RigidSSL-MD arises from the ATLAS dataset, as it offers an exhaustive and non-redundant coverage of the PDB conformational space. Given the gap between computational oracle and real-world wet lab validation, generating a structurally varied set of candidates is especially valuable as it increases the probability of identifying functional hits while avoiding the resource-intensive failure mode of testing redundant, non-viable designs. Furthermore, we examine the distribution of secondary structures, as shown in Figure 2. FoldFlow-2 pretrained with RigidSSL-MD exhibits the greatest secondary structure diversity, fol-

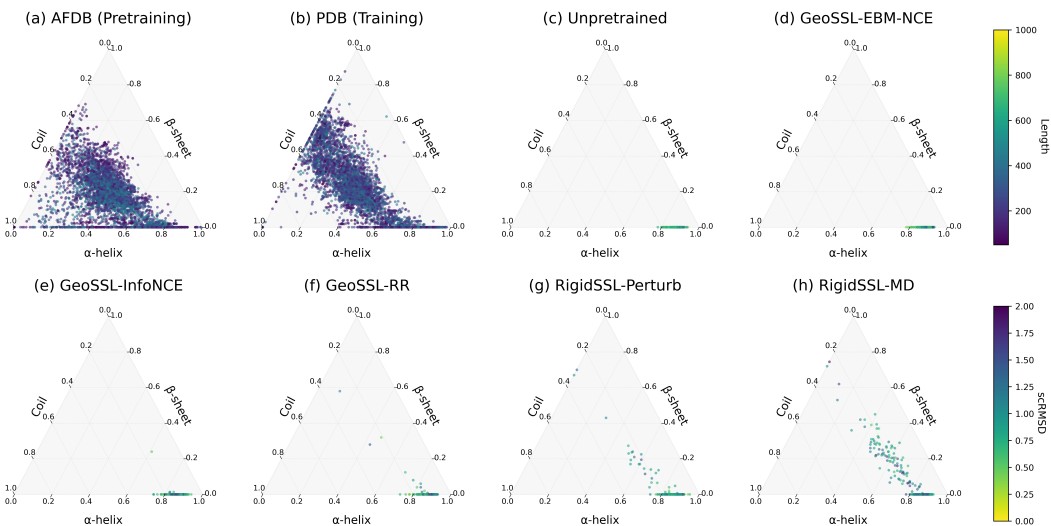

Figure 2: Distribution of secondary structure elements ($\alpha$-helices, $\beta$-sheets, and coils) in protein structure database (a-b) and in designable proteins (scRMSD $\leq 2.0$ Å) generated by FoldFlow-2 under different pretraining methods (c-h). Plots of the structure database are color-coded by sequence length, whereas those of the generated structures are color-coded by scRMSD.

lowed by RigidSSL-Perturb. While the unpretrained model and the GeoSSL-EBM-NCE-pretrained model generate predominantly $\alpha$-helix structures, RigidSSL yields a substantially broader spectrum that includes coils and mixed $\alpha$-helix/$\beta$-sheet compositions.

Next, we quantify alignment between the generated-structure distribution and the reference distributions at both the feature and fold-class levels using the probabilistic metrics of Geffner et al. (2025) (Table 3). FPSD, measured in GearNet (Zhang et al., 2022) feature space, reveals that RigidSSL-MD produces structures whose learned representations most closely match natural protein distributions, achieving the best scores for FrameDiff and second-best for FoldFlow-2 when compared against both PDB and AFDB. This suggests that incorporating structural dynamics encourages the model to generate samples that occupy similar regions of the protein structure manifold as natural proteins. Interestingly, at the topology level, while RigidSSL-MD achieves the best fJSD across both architectures (4.99 for FrameDiff, 5.07 for FoldFlow-2 against PDB), its fS scores (5.35 and 5.54) are lower than those of GeoSSL-InfoNCE (6.86 and 6.01). This indicates that RigidSSL-MD generates structurally diverse proteins whose collective fold distribution closely matches the reference, but individual samples may explore a broader range of conformations within each fold class rather than converging to canonical representatives, leading to softer classifier predictions and consequently lower fold scores.

Furthermore, we conducted a case study on the generation of long protein chains to explore the full potential of RigidSSL in unconditional generation. In this setting, RigidSSL-Perturb enables the model to generate ultra-long proteins of 700–800 residues that remain stereochemically accurate and self-consistent, achieving the best MolProbity score and Clashscore among all pretraining methods. This demonstrates that RigidSSL effectively captures global structural patterns that enable stable and physically realistic long-chain generation. Details are provided in Appendix F.5.

## 4.2 CASE STUDY: ZERO-SHOT MOTIF SCAFFOLDING

Motif scaffolding refers to designing a protein structure that supports a functional motif with fixed geometry. Following Wang et al. (2022a), we formulate this as an "inpainting" task: the model must generate a viable scaffold backbone given only the motif coordinates, without prespecifying the global topology.

We adapted the benchmark suite from Watson et al. (2023), which includes 22 motif-scaffolding targets, in a zero-shot setting, i.e., without task-specific training. Detailed evaluation metrics and results are provided in Appendix F.6. RigidSSL-Perturb achieved the highest average success rate of

15.19%, demonstrating a significant improvement over the unpretrained model (9.35%). Notably, RigidSSL-Perturb exhibited superior robustness on difficult targets that demand longer or more extended scaffolds. For example, on the `5TRV_long` target, RigidSSL-Perturb achieved a success rate of 51%, outperforming the next best method, GeoSSL-InfoNCE, by 21%. While RigidSSL-MD achieved a lower average performance (10.08%) than the perturbation variant, it demonstrated distinct inductive biases, yielding best performance on specific targets such as `2KL8` and `il7ra_gc`.

### 4.3 CASE STUDY: GPCR CONFORMATIONAL ENSEMBLE GENERATION

AlphaFlow (Jing et al., 2024) introduces a novel approach to ensemble generation by re-engineering AlphaFold2 into a flow-matching-based generative model. Instead of predicting a single structure, AlphaFlow learns a continuous flow field that transforms noisy protein conformations into realistic structures, allowing it to sample from the equilibrium ensemble distribution. In particular, AlphaFlow incorporates MD data by finetuning on ensembles of protein conformations sampled from MD trajectories, where each frame represents a near-equilibrium structure along the protein's dynamic landscape. This enables the network to learn how proteins fluctuate across their conformational landscape.

To investigate how RigidSSL's geometric pretraining can affect this process, we selected GPCRs as a representative target. They are particularly difficult to model because their activation involves multiple metastable states connected by slow, collective rearrangements of transmembrane helices, driven by subtle allosteric couplings. This makes their conformational landscape highly anisotropic and multimodal, posing a stringent test for generative models of protein dynamics (Aranda-García et al., 2025). Training and evaluation details can be found in Appendix F.7.1.

We evaluate the generated ensembles across three tiers of increasing biophysical complexity: predicting flexibility, distributional accuracy, and ensemble observables. As shown in Table 6, RigidSSL fundamentally reorganizes the unpretrained AlphaFlow's learned conformational manifold for GPCRs. For flexibility prediction, RigidSSL-Perturb reduces spurious flexibility and yields the closest match to the aggregate diversity of the GPCR MD ensembles. It generates conformations with a pairwise RMSD (2.20) and all-atom RMSF (1.08) that best approximate ground truth targets (1.55 and 1.0, respectively), outperforming the unpretrained model and all pretraining baselines. In terms of distributional accuracy, although GeoSSL variants achieve marginally lower overall root mean $W_2$ and MD PCA $W_2$ distances, RigidSSL-Perturb minimizes the Joint PCA $W_2$-distance (17.53). Crucially, both RigidSSL variants increase the geometric alignment of generated motion modes with MD trajectories, jointly achieving the highest success rate (5.97%) for $> 0.5$ principal component cosine similarity. Finally, we assess complex ensemble observables. This represents the most stringent evaluation category, requiring accurate modeling of sidechain fluctuations, transient structural associations, and higher-order thermodynamic properties. By explicitly incorporating MD dynamics, RigidSSL-MD outperforms all baselines in this regime. It achieves the highest Jaccard similarity for weak contacts (0.43) and cryptically exposed residues (0.71), alongside the highest Spearman correlation ($\rho = 0.03$) for the solvent exposure mutual information matrix. Ultimately, RigidSSL-Perturb primarily enforces collective mode coherence as a structural regularizer, while RigidSSL-MD successfully captures higher-order biophysical statistics to produce realistic contact and exposure profiles.

## 5 DISCUSSION

**RigidSSL-Perturb Improves Geometric Quality of Learned Structures.** In the self-supervised pretraining step in Phase I, we employ data augmentation by adding noise to the original geometry to construct a perturbed geometry as the second view. This choice is motivated by three considerations. (1) Gaussian perturbation can be interpreted as a form of "masking" applied to atom positions in 3D Euclidean space. Analogous to how self-supervised learning masks image patches in computer vision or tokens in NLP, noise injection in geometry partially obscures atomic coordinates and encourages the model to infer consistent global features across corrupted inputs. (2) At small noise scales, this augmentation mimics systematic errors that arise during data collection or prediction. Protein structures from crystallography, cryo-EM, or computational databases such as AFDB often contain small uncertainties in atom placement. Exposing the model to controlled perturbations helps it become robust to such imperfections. (3) Gaussian perturbations can also approximate natural conformational fluctuations around a stable equilibrium. Proteins are dynamic molecules, and their

functional states often involve small deviations from the ground-state structure. Training with noisy perturbations introduces the model to this variability, allowing it to capture flexible yet physically plausible structural patterns.

As shown in Table 1, Phase I pretraining improves designability, indicating that the model's outputs shift toward a more reliable distribution, but this comes at the cost of reduced diversity compared to Phase II. By maximizing mutual information between noisy views, the model learns representations that emphasize stable, fold-defining features while downweighting minor structural deviations. Downstream generators built on such representations therefore tend to produce more reliable and foldable backbones. However, this objective can also suppress conformational variability by collapsing nearby structures into a single representation, thereby limiting the diversity of generated folds.

**RigidSSL-MD Promotes Structural Diversity and Biophysical Fidelity.** In the second phase, RigidSSL-MD incorporates MD trajectories to build augmented views. While this improves structural diversity, it reduces designability compared to perturbation-based pretraining (Table 1). The trade-off arises because MD pretraining is not purely self-supervised in the sense that training views are drawn from force-field-generated trajectories, inheriting biases that may misalign with *de novo* design goals and even cause negative transfer, as seen in prior studies (Liu et al., 2023a). In contrast, the same physically grounded bias improves higher-order biophysical properties (Table 6). Another explanation for the lower designability is that most structure prediction models are trained on stable, static conformations. While MD trajectories capture conformational dynamics to enrich structural flexibility, they also bias the model toward generating metastable conformations rather than the absolute ground states expected by folding and inverse-folding pipelines, thereby hindering re-folding performance. Overall, RigidSSL-MD emphasizes diversity and physical fidelity, whereas RigidSSL-Perturb favors geometric quality described by the designability metric. These two paradigms may therefore serve as complementary strategies, depending on the downstream design objectives.

## ACKNOWLEDGEMENTS

We are grateful to Travis Wheeler and Ken Youens-Clark for maintaining MDRepo, and to the anonymous ICLR reviewers for helpful feedback. We thank the International Max Planck Research School for Intelligent Systems (IMPRS-IS) for supporting Zeju Qiu.

## USE OF LARGE LANGUAGE MODELS

In this work, we used large language models (*e.g.*, ChatGPT) to refine our writing, assist in formulating LaTeX equations, and identify relevant literature.

## ETHICS STATEMENT

This research fully adheres to the ICLR Code of Ethics. The study does not involve human subjects or the use of personal or sensitive data. All datasets and code utilized and released conform to their respective licenses and terms of use. The contributions in this work are foundational and do not raise issues related to fairness, privacy, security, or potential misuse. We confirm that all ethical considerations have been thoroughly addressed.

## REPRODUCIBILITY STATEMENT

We are committed to making our work easy to reproduce. All necessary details for replicating our main experimental results, including data access, experimental setup, model configurations, evaluation metrics, and checkpoints, are publicly available on this repository. Users can follow our documentation and scripts to accurately reproduce the results, ensuring transparency and scientific rigor.

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

# Appendix

## Table of Contents

## A    RELATED WORKS

**Geometric Representation for Small Molecules and Proteins.** A wide range of representation learning models with invariance or equivariance properties have been proposed for both small molecules (Coors et al., 2018; Gasteiger et al., 2020; Schütt et al., 2018) and proteins (Fan et al., 2022; Jing et al., 2020; Wang et al., 2022b), targeting tasks such as property prediction and molecular generation. As a comprehensive benchmark, Geom3D (Liu et al., 2023a) offers a systematic roadmap for geometric representations of small molecules and proteins. It categorizes mainstream approaches into three major families: SE(3)-invariant models, SE(3)-equivariant models with spherical frame bases, and SE(3)-equivariant models with vector frame bases.

**Geometric Pretraining for Small Molecules.** The GraphMVP (Liu et al., 2021) proposes one contrastive objective (EBM-NCE) and one generative objective (variational representation reconstruction, VRR) to optimize the mutual information between the topological and conformational modalities. 3D InfoMax (Stärk et al., 2022) is a special case of GraphMVP, where only the contrastive loss is considered. GeoSSL (Liu et al., 2022a) proposes maximizing the mutual information between noised conformations using an SE(3)-invariant denoising score matching, and a parallel work (Zaidi et al., 2022) is a special case of GeoSSL using only one denoising layer. 3D-EMGP (Jiao et al., 2022) is a parallel work, yet it is E(3)-equivariant, which needlessly satisfies the reflection-equivariant constraint in molecular conformation distribution.

**Geometric Pretraining for Proteins.** Several parallel efforts have been made toward geometric pretraining for proteins. Guo et al. (2022) focuses on maximizing mutual information between diffusion trajectory representations of structurally correlated conformers. GearNet (Zhang et al., 2022) introduces a contrastive self-supervised framework trained on AFDB v1, aiming to maximize the similarity between representations of substructures within the same protein through sparse edge message passing. Similarly, ProteinContrast (Hermosilla & Ropinski, 2022) proposes a contrastive learning approach trained on 476K PDB structures to learn effective 3D protein representations. More recently, MSAGPT (Chen et al., 2024) presents a novel pretraining strategy based on Multiple Sequence Alignments (MSAs), where the model learns to generate MSAs by capturing their statistical distribution, thereby improving structure prediction for proteins with few or no known homologs.

**Binding Pretraining on Stable Positions.** In addition to pretraining on proteins and small molecules separately, several studies have explored joint pretraining strategies that model protein-ligand interactions directly. CoSP (Gao et al., 2022) introduces a co-supervised framework that simultaneously learns 3D representations of protein pockets and ligands using a gated geometric message passing network, incorporating contrastive loss to align real pocket-ligand pairs and employing a chemical similarity-enhanced negative sampling strategy to improve performance. MBP (Yan et al., 2023) addresses the scarcity of high-quality 3D structural data by constructing a pretraining dataset called ChEMBL-Dock, including over 300K protein-ligand pairs and approximately 2.8 million docked 3D structures; MBP employs a multi-task pretraining approach, treating different affinity measurement types (e.g., IC50, Ki, Kd) as separate tasks and incorporating pairwise ranking within the same bioassay to mitigate label noise, thereby learning robust and transferable structural knowledge for binding affinity prediction.

## B    GROUP SYMMETRY AND EQUIVARIANCE

In this article, a 3D molecular graph is represented by a 3D point cloud. The corresponding symmetry group is SE(3), which consists of translations and rotations. Recall that we define the notion of equivariance functions in $\mathbf{R}^3$ in the main text through group actions. Formally, the group SE(3) is said to act on $\mathbf{R}^3$ if there is a mapping $\phi : \text{SE}(3) \times \mathbf{R}^3 \to \mathbf{R}^3$ satisfying the following two conditions:

1. if $e \in \text{SE}(3)$ is the identity element, then
$$\phi(e, \boldsymbol{r}) = \boldsymbol{r} \quad \text{for } \forall \boldsymbol{r} \in \mathbf{R}^3.$$

2. if $g_1, g_2 \in \text{SE}(3)$, then
$$\phi(g_1, \phi(g_2, \boldsymbol{r})) = \phi(g_1 g_2, \boldsymbol{r}) \quad \text{for } \forall \boldsymbol{r} \in \mathbf{R}^3.$$

Then, there is a natural SE(3) action on vectors $\boldsymbol{r}$ in $\mathbf{R}^3$ by translating $\boldsymbol{r}$ and rotating $\boldsymbol{r}$ for multiple times. For $g \in \text{SE}(3)$ and $\boldsymbol{r} \in \mathbf{R}^3$, we denote this action by $g\boldsymbol{r}$. Once the notion of group action is

defined, we say a function $f : \mathbf{R}^3 \to \mathbf{R}^3$ that transforms $\boldsymbol{r} \in \mathbf{R}^3$ is equivariant if:

$$f(g\boldsymbol{r}) = gf(\boldsymbol{r}), \quad \text{for } \forall \ \boldsymbol{r} \in \mathbf{R}^3.$$

On the other hand, $f : \mathbf{R}^3 \to \mathbf{R}^1$ is invariant, if $f$ is independent of the group actions:

$$f(g\boldsymbol{r}) = f(\boldsymbol{r}), \quad \text{for } \forall \ \boldsymbol{r} \in \mathbf{R}^3.$$

## C  PROTEIN BACKBONE PARAMETERIZATION AND CANONICALIZATION

### C.1  LOCAL FRAME CONSTRUCTION

Following AlphaFold2 (Jumper et al., 2021), we construct local coordinate frames from atomic coordinates in protein structures. Each residue frame is defined as a rigid transformation $(\vec{t}, r)$, where $\vec{t} \in \mathbb{R}^3$ is the translation vector and $r \in \mathrm{SO}(3)$ is the rotation matrix. Frames are derived from the backbone atoms N, $C_\alpha$, and C, with coordinates $x_1$, $x_2$, and $x_3$, respectively. The translation vector $\vec{t}$ is set to the $C_\alpha$ position $x_2$, while the rotation matrix $r$ is computed using Gram–Schmidt orthogonalization in Algorithm 1. As a result, each residue $i$ in a protein $g = \{(\vec{t}_i, r_i)\}_{i=1}^L$ is thus associated with an orthonormal frame centered at its backbone atom.

---
**Algorithm 1** Frame construction via Gram–Schmidt orthogonalization
---
1: **Input:** Three atomic positions $x_1, x_2, x_3 \in \mathbb{R}^3$
2: **Output:** Translation vector $\vec{t} \in \mathbb{R}^3$ and rotation matrix $r \in \mathbb{R}^{3 \times 3}$
3: $\vec{v}_1 \leftarrow x_3 - x_2$
4: $\vec{v}_2 \leftarrow x_1 - x_2$
5: $\vec{e}_1 \leftarrow \vec{v}_1 / \|\vec{v}_1\|$
6: $\vec{u}_2 \leftarrow \vec{v}_2 - (\vec{e}_1^\top \vec{v}_2)\, \vec{e}_1$
7: $\vec{e}_2 \leftarrow \vec{u}_2 / \|\vec{u}_2\|$
8: $\vec{e}_3 \leftarrow \vec{e}_1 \times \vec{e}_2$
9: $r \leftarrow [\vec{e}_1, \vec{e}_2, \vec{e}_3]$
10: $\vec{t} \leftarrow x_2$
11: **return** $(\vec{t}, r)$

---

### C.2  CANONICALIZATION DETAILS

The eigendecomposition in Section 3.1 requires two additional steps to yield a deterministic, valid rotation matrix $\mathbf{V} \in \mathrm{SO}(3)$.

**Axis ordering.** We sort the eigenvectors by their corresponding eigenvalues in ascending order, establishing a deterministic ordering of the principal axes.

**Right-handedness.** Eigenvectors are only guaranteed to form an orthogonal basis ($\mathbf{V} \in O(3)$), and may include a reflection when $\det(\mathbf{V}) = -1$. We enforce $\mathbf{V} \in \mathrm{SO}(3)$ by flipping one axis via $\mathbf{V} \leftarrow \mathbf{V} \, \mathrm{diag}(1, 1, \det(\mathbf{V}))$, which negates the third eigenvector when $\det(\mathbf{V}) = -1$.

## D  ROTATION PERTURBATION IN RIGIDSSL-PERTURB

Recall that a structure $g$ is modeled as a sequence of $L$ rigid residues, with each residue represented by a translation and rotation transformation: $g = \{(\vec{t}_i, r_i)\}_{i=1}^L$. In RigidSSL-Perturb, we independently perturb each residue's translation $\vec{t}_i$ and rotation $r_i$. This section details the mathematical foundation, implementation, and implications of rotation perturbation.

### D.1  PARAMETRIZATIONS OF $\mathrm{SO}(3)$

The rotation group $\mathrm{SO}(3)$ admits several equivalent representations. We review its group structure, Lie algebra, axis-angle parameterization, and the exponential and logarithm maps, which together provide the foundations for defining and sampling rotational perturbations in RigidSSL-Perturb.

### D.1.1 ROTATION GROUP AND MANIFOLD STRUCTURE

The special orthogonal group $SO(3) = \{R \in \mathbb{R}^{3\times3} \mid R^T R = I, \det(R) = 1\}$ forms a 3-dimensional compact Lie group manifold representing all possible rotations in 3D space. As a manifold, $SO(3)$ is diffeomorphic to real projective space $\mathbb{RP}^3$, obtained by identifying antipodal points on the 3-sphere $S^3$.

### D.1.2 LIE ALGEBRA AND TANGENT SPACE

The Lie algebra $\mathfrak{so}(3)$ is the tangent space to $SO(3)$ at the identity matrix $I$. It consists of all $3 \times 3$ skew-symmetric matrices: $\mathfrak{so}(3) = \{X \in \mathbb{R}^{3\times3} \mid X^T = -X\}$. Elements of $\mathfrak{so}(3)$ can be interpreted as infinitesimal rotations or angular velocities. A remarkable isomorphism exists between $\mathfrak{so}(3)$ and $\mathbb{R}^3$ via the "hat map": $\wedge : \mathbb{R}^3 \to \mathfrak{so}(3), \quad \boldsymbol{\omega}^\wedge = \begin{pmatrix} 0 & -\boldsymbol{\omega}_3 & \boldsymbol{\omega}_2 \\ \boldsymbol{\omega}_3 & 0 & -\boldsymbol{\omega}_1 \\ -\boldsymbol{\omega}_2 & \boldsymbol{\omega}_1 & 0 \end{pmatrix}$. The inverse "vee map" is denoted $\vee : \mathfrak{so}(3) \to \mathbb{R}^3$.

### D.1.3 AXIS-ANGLE REPRESENTATION OF ROTATION

Axis-angle rotation representations exist in two related forms:

**Axis-angle representation.** A *geometric* pair $(\mathbf{n}, \theta)$ where $\mathbf{n} \in S^2$ is a unit vector representing the rotation axis and $\theta \in [0, \pi]$ is the rotation angle. This representation lives in the product space $S^2 \times [0, \pi]$ with identifications: $(\mathbf{n}, 0) \sim (-\mathbf{n}, 0)$ and $(\mathbf{n}, \pi) \sim (-\mathbf{n}, \pi)$.

**Axis-angle vector (rotation vector).** An *algebraic* vector $\boldsymbol{\omega} = \theta \mathbf{n} \in \mathbb{R}^3$ where the direction specifies the rotation axis and the magnitude $|\boldsymbol{\omega}| = \theta$ specifies the rotation angle. This representation belongs to a ball of radius $\pi$ in $\mathbb{R}^3$ with antipodal points on the boundary identified. The axis-angle vector is precisely the exponential coordinates in $\mathfrak{so}(3)$ under the isomorphism provided by the hat map. That is, for a rotation vector $\boldsymbol{\omega}$, the corresponding element in $\mathfrak{so}(3)$ is $\boldsymbol{\omega}^\wedge$.

### D.1.4 EXPONENTIAL MAP

The exponential map $\exp : \mathfrak{so}(3) \to SO(3)$ connects the Lie algebra to the Lie group: $\exp(X) = \sum_{k=0}^{\infty} \frac{X^k}{k!}$ For an axis-angle vector $\boldsymbol{\omega} = \theta\mathbf{n}$, Rodrigues' formula provides a closed-form expression for the corresponding rotation matrix: $\exp(\boldsymbol{\omega}^\wedge) = I + \frac{\sin\theta}{\theta}\boldsymbol{\omega}^\wedge + \frac{1-\cos\theta}{\theta^2}(\boldsymbol{\omega}^\wedge)^2$. When $\theta = 0$, this reduces to the identity matrix $I$. The exponential map is surjective but not injective globally and wraps around when $|\boldsymbol{\omega}| > \pi$.

### D.1.5 LOGARITHM MAP

The logarithm map $\log : SO(3) \to \mathfrak{so}(3)$ is the (local) inverse of the exponential map. For a rotation matrix $R \in SO(3)$, where $\text{trace}(R) \neq -1$: $\log(R) = \frac{\theta}{2\sin\theta}(R - R^T)$ where $\theta = \cos^{-1}\left(\frac{\text{trace}(R)-1}{2}\right)$. The logarithm map yields an axis-angle vector representation when composed with the vee map: $\boldsymbol{\omega} = (\log(R))^\vee$.

### D.2 ISOTROPIC GAUSSIAN DISTRIBUTION ON $SO(3)$: $\mathcal{IG}_{SO(3)}$

With axis-angle representation of rotations, $\mathcal{IG}_{SO(3)}$ consists of two components:

**Axis Distribution.** The axis distribution is uniform over the unit sphere $S^2$, with PDF: $p(\hat{\mathbf{n}}) = \frac{1}{4\pi}$, where $\hat{\mathbf{n}} \in S^2$ is a unit vector representing the axis of rotation.

**Angle Distribution.** The angle distribution for $\theta \in [0, \pi]$ has the probability density function:

$$p(\theta) = \frac{1 - \cos\theta}{\pi} \sum_{l=0}^{\infty} (2l+1) e^{-l(l+1)\epsilon^2} \frac{\sin((l + \frac{1}{2})\theta)}{\sin(\frac{\theta}{2})}, \tag{13}$$

where $\epsilon$ is the concentration parameter of $\mathcal{IG}_{\mathrm{SO}(3)}$. For small values of $\epsilon$ where numerical convergence of the infinite sum becomes difficult, an approximation can be used:

$$p(\theta) = \frac{(1 - \cos(\theta))}{\pi} \sqrt{\pi} \epsilon^{-\frac{3}{2}} e^{\frac{\epsilon}{4}} e^{-\frac{(\frac{\theta}{2})^2}{\epsilon}} \cdot \frac{\left[ \theta - e^{-\frac{\pi^2}{\epsilon}} \left( (\theta - 2\pi) e^{\frac{\pi\theta}{\epsilon}} + (\theta + 2\pi) e^{-\frac{\pi\theta}{\epsilon}} \right) \right]}{2 \sin\left(\frac{\theta}{2}\right)}. \quad (14)$$

As $\epsilon \to \infty$, $p(\theta)$ approaches the uniform distribution on SO(3):

$$f_{\mathrm{uniform}}(\theta) = \frac{1 - \cos\theta}{\pi}. \quad (15)$$

### D.3 ROTATION SAMPLING FROM $\mathcal{IG}_{\mathrm{SO}(3)}$

A sample from $\mathcal{IG}_{\mathrm{SO}(3)}(\mu, \epsilon^2)$ can be decomposed as $r = \mu r_c$ where $r_c \sim \mathcal{IG}_{\mathrm{SO}(3)}(I, \epsilon^2)$. Therefore, we first sample from an identity-centered distribution and then rotate by $\mu$. Specifically, the sampling process is as follows:

**Sample the rotation axis.** Generate a random unit vector $\hat{n} \in S^2$ uniformly by sampling a point $(x, y, z)$ from the standard normal distribution $\mathcal{N}(0, I_3)$ and normalizing to unit length.

**Sample the rotation angle.** Sample $\theta \in [0, \pi]$ from the density $p(\theta)$ described above using inverse transform sampling. Specifically, numerically compute the CDF $F(\theta) = \int_0^\theta p(t)\, dt$, sample $u \sim \mathrm{Uniform}[0, 1]$, and compute $\theta = F^{-1}(u)$ through numerical interpolation.

**Construct the exponential coordinates.** Form the axis-angle vector $\boldsymbol{\omega} = \theta \hat{n} \in \mathbb{R}^3$. This vector represents elements in the Lie algebra $\mathfrak{so}(3)$ via the hat map: $\boldsymbol{\omega}^\wedge \in \mathfrak{so}(3)$, as detailed in Appendix D.1.

**Apply the exponential map.** Convert $\boldsymbol{\omega}$ to a rotation matrix using Rodrigues' formula:

$$r_c = \exp(\boldsymbol{\omega}^\wedge) = I + \frac{\sin\theta}{\theta} \boldsymbol{\omega}^\wedge + \frac{1 - \cos\theta}{\theta^2} (\boldsymbol{\omega}^\wedge)^2. \quad (16)$$

**Apply the mean rotation.** Compute the final rotation as:

$$r = \mu r_c. \quad (17)$$

### D.4 ROTATION PERTURBATION EFFECTS IN $\mathbb{R}^3$

When applying rotational perturbations sampled from $\mathcal{IG}_{\mathrm{SO}(3)}$, the geometric effect in $\mathbb{R}^3$ varies with initial rotation despite consistent geodesic distances on SO(3). For rotations $R_a^0, R_b^0 \in \mathrm{SO}(3)$ and point $p \in \mathbb{R}^3$, with perturbations of equal magnitude:

$$R_a^1 = R_a^0 \cdot \exp(\boldsymbol{\omega}_a^\wedge), \quad R_b^1 = R_b^0 \cdot \exp(\boldsymbol{\omega}_b^\wedge), \quad (18)$$

where $\boldsymbol{\omega}_a, \boldsymbol{\omega}_b \in \mathbb{R}^3$ are axis-angle vectors with $\|\boldsymbol{\omega}_a\| = \|\boldsymbol{\omega}_b\| = \theta$. The resulting displacements

$$\Delta_a = R_a^1 p - R_a^0 p = R_a^0 \left( \exp(\boldsymbol{\omega}_a^\wedge) - \mathbf{I} \right) p, \quad (19)$$

$$\Delta_b = R_b^1 p - R_b^0 p = R_b^0 \left( \exp(\boldsymbol{\omega}_b^\wedge) - \mathbf{I} \right) p \quad (20)$$

generally have $|\Delta_a| \neq |\Delta_b|$, as displacement magnitude depends on the initial rotation and point position.

## E  BASE PROTEIN ENCODER

We adopt the IPA module from AlphaFold2 (Jumper et al., 2021) as the base protein encoder. Specifically, the edge embedding is initialized using a distogram, which discretizes the continuous pairwise distances into a one-hot vector representation. For any two residues $i$ and $j$, the Euclidean distance $d_{ij} = \|\mathbf{x}_i - \mathbf{x}_j\|_2$ is categorized into one of $N_{\mathrm{bins}}$ discrete bins. The $k$-th component of the resulting embedding vector is determined by the indicator function $\mathbb{I}(\cdot)$:

$$\mathrm{EdgeEmbed}(d_{ij})_k = \mathbb{I}(l_k < d_{ij} < u_k), \quad \text{for } k = 1, \dots, N_{\mathrm{bins}}, \quad (21)$$

where $(l_k, u_k)$ are the lower and upper bounds of the $k$-th distance bin. In our model, we set $N_{\text{bins}} = 22$, with bins linearly spaced between a minimum of $1 \times 10^{-5}$ Å and a maximum of 20.0 Å. The final bin extends to infinity to capture all larger distances.

Next, like normal attention mechanisms, we generate the scalar and point-based query, key, and value features for each node using bias-free linear transformations:

$$\mathbf{q}_i^h, \mathbf{k}_i^h, \mathbf{v}_i^h = \mathbf{W}_1 \cdot \mathbf{s}_i, \tag{22}$$

$$\mathbf{q}_i^{hp}, \mathbf{k}_i^{hp}, \mathbf{v}_i^{hp} = \mathbf{W}_2 \cdot \mathbf{s}_i, \tag{23}$$

$$\mathbf{b}_{ij}^h = \mathbf{W}_3 \cdot \mathbf{z}_{ij}, \tag{24}$$

where $\mathbf{W}_1 \in \mathbb{R}^{(h \cdot d) \times d_s}$, $\mathbf{W}_2 \in \mathbb{R}^{(h \cdot p \cdot 3) \times d_s}$, $\mathbf{W}_3 \in \mathbb{R}^{h \times d_z}$, and $\mathbf{s}_i \in \mathbb{R}^{d_{\text{in}}}$ is the input node embedding. Here, $h$ denotes the number of attention heads, and $p$ is the number of reference points used per head in the 3D attention mechanism. Each point-based feature (e.g., $\mathbf{q}_i^{hp}$) is reshaped into $[h, p, 3]$, representing a set of 3D vectors per head that enable spatial reasoning in the IPA module.

A key component then, is calculating the attention score:

$$a_{ij}^h = \text{softmax}_j \left( w_L \left[ \frac{\mathbf{q}_i^\top \mathbf{k}_j^h}{\sqrt{d}} + b_{ij}^h - \frac{\gamma^h w_C}{2} \sum_p \left\| T_i \circ \mathbf{q}_i^{hp} - T_j \circ \mathbf{k}_j^{hp} \right\|^2 \right] \right), \tag{25}$$

where $T \circ y = (r \cdot y + x)$ represents the transformation.

As shown in Equation (25), IPA introduces an additional bias term besides the bias $b_{ij}$ from the pairwise feature, $\sum_p \left\| T_i \circ \mathbf{q}_i^{hp} - T_j \circ \mathbf{k}_j^{hp} \right\|^2$, which explicitly models the spatial relationship between nodes after applying the local transformations $T_i$ and $T_j$. When the transformed point $T_i \circ \mathbf{q}_i^{hp}$ and $T_j \circ \mathbf{k}_j^{hp}$ differ significantly, it indicates a large spatial discrepancy between node $i$ and node $j$, and the resulting attention score is correspondingly reduced.

We then calculate the attention score and update the node representation:

$$\tilde{\mathbf{o}}_i^h = \sum_j a_{ij}^h \mathbf{z}_{ij}, \tag{26}$$

$$\mathbf{o}_i^h = \sum_j a_{ij}^h \mathbf{v}_j^h, \tag{27}$$

$$\tilde{\mathbf{o}}_i^{hp} = T_i^{-1} \circ \left( \sum_j a_{ij}^h \left( T_j \circ \tilde{\mathbf{v}}_j^{hp} \right) \right), \tag{28}$$

$$\tilde{\mathbf{s}}_i = \text{Linear} \left( \text{concat}_{h,p} \left( \tilde{\mathbf{o}}_i^h, \mathbf{o}_i^h, \tilde{\mathbf{o}}_i^{hp}, \|\tilde{\mathbf{o}}_i^{hp}\| \right) \right). \tag{29}$$

In $\tilde{\mathbf{o}}_i^{hp} = T_i^{-1} \circ \left( \sum_j a_{ij}^h \left( T_j \circ \tilde{\mathbf{v}}_j^{hp} \right) \right)$, the model first applies the global transformation $T_j$ to each neighboring point vector $\tilde{\mathbf{v}}_j^{hp}$, weighs them by the attention scores $a_{ij}^h$, and then aggregates the results in the global frame. The result is then transformed back to the local coordinate frame of node $i$ using $T_i^{-1}$. This ensures that the update is the same in the local frame though a global transform is applied.

To make it compatible with our flow matching setting, after each IPA block, we embed the current time step $t$ using sinusoidal positional embedding (Ho et al., 2020) denoted as $F_t$ and add that to $\tilde{\mathbf{s}}_i$ to complete the full update:

$$\tilde{\mathbf{s}}_i = \tilde{\mathbf{s}}_i + F_t(t). \tag{30}$$

Finally, we map the updated node representation back to the translation and quaternion:

$$x_i, q_i = Linear(\tilde{\mathbf{s}}_i), \tag{31}$$

where $x_i$ and $q_i$ represent the translation and rotation (quaternion) respectively.

# F  EXPERIMENT DETAILS

## F.1  TRAINING OVERVIEW

Table 2 summarizes the training setups for our method and all baselines, including data sources, dataset sizes, protein length ranges, training objectives, and computational resources. Additional implementation and training details are provided in the subsequent sections.

Table 2: Overview of pretraining and downstream training configurations.

| Method | Stage | Data Source | Data Size | Length | Objective | Compute & Time |
|---|---|---|---|---|---|---|
| GeoSSL-InfoNCE | Pretraining | AFDB (UniProtKB) | 432,194 | $60-512$ | Contrastive (InfoNCE) | 4 days ($4\times$A100) |
| GeoSSL-EBM-NCE | Pretraining | AFDB (UniProtKB) | 432,194 | $60-512$ | Contrastive (EBM-NCE) | 4 days ($4\times$A100) |
| GeoSSL-RR | Pretraining | AFDB (UniProtKB) | 432,194 | $60-512$ | Generative (Representation Reconstruction) | 3 days ($4\times$A100) |
| RigidSSL-Perturb (Ours) | Pretraining (Phase I) | AFDB (UniProtKB) | 432,194 | $60-512$ | Generative (RigidSSL) | 2.75 days ($1\times$H100) |
| RigidSSL-MD (Ours) | Pretraining (Phase II) | ATLAS | 1,390 traj. | $60-512$ | Generative (RigidSSL) | 1.88 days ($1\times$H100) |
| FrameDiff | Finetuning | PDB | 20,312 | $60-512$ | SE(3) Diffusion | 7 days ($4\times$H100) |
| FoldFlow-2 | Finetuning | PDB & AFDB | 20,312 | $60-384$ | SE(3) Flow Matching | 4 days ($2\times$A100) |

## F.2  PRETRAINING: DATASET

**RigidSSL-Perturb.** For the first phase of our pretraining strategy, we use version 4 of AFDB. Specifically, we utilize AFDB's coverage for the UniProtKB/Swiss-Prot section of the UniProt KnowledgeBase (UniProtKB), which comprises 542,378 proteins. We filter sequences by length, retaining only those between 60 and 512 residues, yielding 432,194 proteins for training.

To determine the optimal translation and rotation noise levels for constructing perturbed views in RigidSSL-Perturb, we conducted an ablation study (see Appendix G) and selected a translation noise scale of $\sigma = 0.03$ and a rotation noise scale of $\epsilon = 0.5$.

**RigidSSL-MD.** For the second pretraining phase, we utilize the ATLAS dataset, obtained from MDRepo (Roy et al., 2025; Vander Meersche et al., 2024). This dataset contains 100 ns all-atom MD simulation triplicates for 1,390 protein chains, which were selected to provide an exhaustive and non-redundant sampling of the PDB's conformational space based on the Evolutionary Classification of Domains (ECOD) (Cheng et al., 2014). All simulations were performed with GROMACS using the CHARMM36m force field; each protein was solvated in a triclinic box with TIP3P water and neutralized with $Na^+/Cl^-$ ions at a 150 $mM$ concentration. We filter sequences by length, retaining only those between 60 and 512 residues. To generate training samples for RigidSSL-MD, we extracted 80 pairs of adjacent conformations separated by a 2 ns interval from each trajectory to reflect the conformational fluctuations of the protein structures.

## F.3  PRETRAINING: OPTIMIZATION

For training RigidSSL-Perturb and RigidSSL-MD, we use Adam optimizer (Kingma & Ba, 2015) with learning rate of 0.0001. RigidSSL-Perturb was trained for 2.75 days on one NVIDIA H100 GPU with batch size of 1. RigidSSL-MD was trained for 1.88 days on one NVIDIA H100 GPU with batch size of 1.

## F.4  DOWNSTREAM: UNCONDITIONAL GENERATION

### F.4.1  TRAINING AND INFERENCE FOR UNCONDITIONAL GENERATION

We trained FrameDiff and FoldFlow-2 using the same dataset preprocessing, model architecture, and optimization hyperparameters as in their respective original works (Huguet et al., 2024; Yim et al., 2023), but warm-started the IPA modules with pretrained weights. FrameDiff was trained for one week on four NVIDIA H100 GPUs, whereas FoldFlow-2 was trained for four days on two NVIDIA A100 80 GB GPUs.

For the results in Table 1, we generate structures by sampling backbones of lengths 100–300 residues with FrameDiff and 100–600 residues with FoldFlow-2, in increments of 50, using the inference hyperparameters specified in the original works (Huguet et al., 2024; Yim et al., 2023). For the long-chain case study in Appendix F.5, we follow the same inference procedure but extend the target lengths to 700 and 800 residues. We then select the structure with the highest self-consistency and assess its stereochemical quality using MolProbity (Davis et al., 2007).

### F.4.2 Evaluation Metrics for Unconditional Generation

**Designability** is a self-consistency metric that examines if there exist amino acid sequences that can fold into the generated structure. Specifically, we employ ProteinMPNN (Dauparas et al., 2022) to design sequences (i.e. reverse fold) for FrameDiff/FoldFlow-2 structures, which are then folded with ESMFold (Lin et al., 2023) to compare with the original structure. We quantify self-consistency through scRMSD and employ scRMSD $\leq 2$ Å as the criterion for being designable.

**Novelty** assesses whether the model can generalize beyond the training set and produce novel backbones dissimilar from those in PDB. We use FoldSeek (Van Kempen et al., 2024) to search for similar structures and report the highest TM-scores (measure of similarity between two protein structures) of samples to any chain in PDB.

**Diversity** measures structural differences among the generated samples. We quantify diversity in two ways: (1) the length-averaged mean pairwise TM-score among designable samples, and (2) the number of distinct structural clusters obtained using MaxCluster (Herbert, A. & Sternberg, M., 2008), which hierarchically clusters backbones using average linkage with sequence-independent structural alignment at a TM-score threshold of 0.7. We report MaxCluster diversity as the proportion of unique clusters, i.e., (number of clusters) / (number of samples).

However, these standard protein-structure generation benchmarks only emphasize sample-level criteria but do not directly assess whether a generator matches a target *distribution* of structures. Following Geffner et al. (2025), we additionally report in Table 3 three distribution-level metrics derived from a fold classifier: the Fréchet Protein Structure Distance, the fold Jensen–Shannon divergence, and the fold score.

**Fold classifier and representations.** Let $x$ denote a protein backbone structure. We use a GearNet-based fold classifier $p_\phi(\cdot \mid x)$ (Zhang et al., 2022) that outputs a probability distribution over fold classes. The classifier also induces a non-linear feature extractor $\varphi(x)$ (we use the last-layer features of the classifier), which we treat as an embedding of protein backbones.

Let $\mathcal{X}_{\mathrm{gen}}$ and $\mathcal{X}_{\mathrm{ref}}$ denote sets (or empirical distributions) of generated and reference structures, respectively. Define the empirical feature moments:

$$
\begin{aligned}
\mu_{\mathrm{gen}} = \mathbb{E}_{x\sim\mathcal{X}_{\mathrm{gen}}}[\varphi(x)], \quad & \Sigma_{\mathrm{gen}} = \mathbb{E}_{x\sim\mathcal{X}_{\mathrm{gen}}}\big[(\varphi(x)-\mu_{\mathrm{gen}})(\varphi(x)-\mu_{\mathrm{gen}})^\top\big], \\
\mu_{\mathrm{ref}} = \mathbb{E}_{x\sim\mathcal{X}_{\mathrm{ref}}}[\varphi(x)], \quad & \Sigma_{\mathrm{ref}} = \mathbb{E}_{x\sim\mathcal{X}_{\mathrm{ref}}}\big[(\varphi(x)-\mu_{\mathrm{ref}})(\varphi(x)-\mu_{\mathrm{ref}})^\top\big].
\end{aligned}
\tag{32}
$$

We also define the *marginal* predicted label distributions:

$$
\begin{aligned}
p_{\mathrm{gen}}(\cdot) = \mathbb{E}_{x\sim\mathcal{X}_{\mathrm{gen}}}\big[p_\phi(\cdot \mid x)\big], \\
p_{\mathrm{ref}}(\cdot) = \mathbb{E}_{x\sim\mathcal{X}_{\mathrm{ref}}}\big[p_\phi(\cdot \mid x)\big].
\end{aligned}
\tag{33}
$$

**Fréchet Protein Structure Distance** FPSD compares generated and reference backbones in the non-linear feature space $\varphi(x)$ by approximating each feature distribution as a Gaussian and measuring the Fréchet distance between these Gaussians:

$$
\mathrm{FPSD}(\mathcal{X}_{\mathrm{gen}}, \mathcal{X}_{\mathrm{ref}}) := \|\mu_{\mathrm{gen}} - \mu_{\mathrm{ref}}\|_2^2 + \mathrm{tr}\Big(\Sigma_{\mathrm{gen}} + \Sigma_{\mathrm{ref}} - 2\big(\Sigma_{\mathrm{gen}}\Sigma_{\mathrm{ref}}\big)^{1/2}\Big).
\tag{34}
$$

Lower FPSD indicates closer alignment between generated and reference distributions in feature space.

**Fold Jensen–Shannon divergence** fJSD measures discrepancy in the *categorical* fold-label space by computing the Jensen–Shannon divergence between marginal predicted label distributions:

$$
\mathrm{fJSD}(\mathcal{X}_{\mathrm{gen}}, \mathcal{X}_{\mathrm{ref}}) := 10 \times D_{\mathrm{JS}}\big(p_{\mathrm{gen}} \,\|\, p_{\mathrm{ref}}\big).
\tag{35}
$$

The Jensen–Shannon divergence is

$$
\begin{aligned}
D_{\mathrm{JS}}(P\|Q) := \tfrac{1}{2}D_{\mathrm{KL}}(P\|M) + \tfrac{1}{2}D_{\mathrm{KL}}(Q\|M), \\
M := \tfrac{1}{2}(P + Q).
\end{aligned}
\tag{36}
$$

We multiply by 10 for reporting convenience. Lower fJSD indicates closer agreement of generated vs. reference fold distributions. When fold labels are hierarchical (e.g., C/A/T), fJSD can be computed at each level; we report the average across levels.

**Fold score** The fold score summarizes both per-sample confidence and across-sample diversity under the fold classifier. Let $p_{\text{gen}}(\cdot) = \mathbb{E}_{x \sim \mathcal{X}_{\text{gen}}}[p_\phi(\cdot \mid x)]$ be the marginal label distribution over generated samples. Then

$$\text{fS}(\mathcal{X}_{\text{gen}}) := \exp\Big(\mathbb{E}_{x \sim \mathcal{X}_{\text{gen}}}\big[D_{\text{KL}}(p_\phi(\cdot \mid x) \,\|\, p_{\text{gen}}(\cdot))\big]\Big). \tag{37}$$

A higher fS indicates that individual samples are assigned sharp fold distributions while the aggregate covers a broad range of fold classes.

Table 3: Comparison of FPSD, fS, and fJSD. Lower is better for FPSD/fJSD; higher is better for fS. Best mean values are in **bold** and second-best are underlined.

| Model | Pretraining | FPSD vs. | | fS | fJSD vs. | |
|---|---|---|---|---|---|---|
| | | PDB ($\downarrow$) | AFDB ($\downarrow$) | (C/A/T) ($\uparrow$) | PDB (C/A/T) ($\downarrow$) | AFDB (C/A/T) ($\downarrow$) |
| FrameDiff | None | 908.743 | 840.682 | 1.052/1.827/4.249 | 3.174/3.902/5.454 | 2.209/3.119/4.807 |
| FrameDiff | GeoSSL-EBM-NCE | 827.919 | 770.448 | **1.108/2.243**/6.025 | **3.017**/3.593/5.079 | **2.068**/2.807/4.408 |
| FrameDiff | GeoSSL-InfoNCE | 858.332 | 806.346 | 1.002/2.060/**6.862** | 3.360/3.817/5.195 | 2.369/2.965/4.419 |
| FrameDiff | GeoSSL-RR | 892.657 | 846.222 | 1.002/1.851/6.559 | 3.364/**3.474**/5.037 | 2.373/**2.452**/4.357 |
| FrameDiff | RigidSSL-Perturb (ours) | 901.424 | 843.407 | 1.003/1.735/4.105 | 3.359/3.605/5.417 | 2.370/2.632/4.687 |
| FrameDiff | RigidSSL-MD (ours) | **776.297** | **701.705** | 1.001/1.834/5.345 | 3.374/3.552/**4.987** | 2.383/2.560/**4.151** |
| FoldFlow-2 | None | 1273.363 | 1198.292 | 1.098/1.464/4.855 | 2.813/3.955/5.616 | 1.903/3.184/5.211 |
| FoldFlow-2 | GeoSSL-EBM-NCE | 1250.866 | 1201.031 | 1.041/1.565/5.651 | 3.099/4.107/5.834 | 2.131/3.382/5.570 |
| FoldFlow-2 | GeoSSL-InfoNCE | 1182.036 | 1127.343 | 1.075/1.788/**6.005** | 2.940/3.492/5.566 | 1.996/2.602/5.207 |
| FoldFlow-2 | GeoSSL-RR | 2532.639 | 2485.163 | **1.681/2.497**/4.715 | **1.219**/3.374/5.240 | **0.885**/3.279/5.227 |
| FoldFlow-2 | RigidSSL-Perturb (ours) | **959.982** | **875.549** | 1.015/1.647/4.404 | 3.288/3.921/5.294 | 2.305/3.117/**4.620** |
| FoldFlow-2 | RigidSSL-MD (ours) | 1113.829 | 1070.094 | 1.198/2.249/5.535 | 2.623/**3.185**/5.073 | 1.771/**2.520**/4.810 |

## F.5 CASE STUDY: LONG CHAIN GENERATION

To explore the potential of RigidSSL, we conducted a case study on the generation of long protein (details provided in F.4.1). Whereas the original FoldFlow-2 analysis was restricted to sequences of 100–300 residues, we extend the evaluation to ultra-long sequences of 700 and 800 residues under different pretraining methods. We visualize the best structure in terms of scRMSD among 50 generated samples in Figure 3. Notably, only FoldFlow-2 pretrained with RigidSSL-Perturb is able to produce self-consistent and thus designable structures.

To further understand the stereochemistry of the generated structures, we use MolProbity (Davis et al., 2007) to compute: (1) Clashscore, which is the number of serious steric overlaps ($>0.4$ Å) per 1000 atoms, and (2) the MolProbity score, which is a composite metric integrating Clashscore, rotamer, and Ramachandran evaluations. As shown in Table 4, FoldFlow-2 pretrained with RigidSSL-Perturb achieves the best clashscore and MolProbity score, indicating that it effectively captures the biophysical constraints underlying long, complex protein structures despite being trained only on shorter sequences (60–384 residues). This demonstrates that RigidSSL-Perturb substantially improves the model's ability to maintain stereochemical accuracy at long sequence lengths, suggesting robust learning of global structural patterns that generalize beyond the training regime.

Table 4: Comparison of Clashscore and MolProbity scores for 700- and 800-residue proteins generated by FoldFlow-2 under different pretraining methods. The best values are shown in **bold** and the second-best are underlined.

| Metric | Length | Unpretrained | GeoSSL-EBM-NCE | GeoSSL-RR | GeoSSL-InfoNCE | RigidSSL-Perturb | RigidSSL-MD |
|---|---|---|---|---|---|---|---|
| Clashscore ($\downarrow$) | 700 | 152.53 | 45.11 | 96.38 | 55.02 | **21.36** | 37.16 |
| | 800 | 172.36 | 61.70 | 123.16 | 74.98 | **26.42** | 39.35 |
| MolProbity score ($\downarrow$) | 700 | 2.64 | 2.22 | 2.51 | 2.21 | **1.82** | 2.05 |
| | 800 | 2.69 | 2.38 | 2.61 | 2.34 | **1.91** | 2.08 |

## F.6 CASE STUDY: ZERO-SHOT MOTIF SCAFFOLDING

We evaluate motif scaffolding by adapting the RFdiffusion protocol (Watson et al., 2023). For each target motif, we generate 100 backbone scaffolds using FoldFlow-2 models pretrained under different frameworks and finetuned for unconditional backbone generation. Unlike the approach in Huguet et al. (2024), we did not employ additional finetuning via pseudo-label training. Motif residues are held fixed during generation by setting their corresponding entries in the fixed mask to 1.0, while scaffold residues are sampled from the learned flow. We use 50 inference steps with noise scale 0.1 and a minimum time of $t_{\min} = 0.01$. The motif structure is incorporated as a conditioning signal by

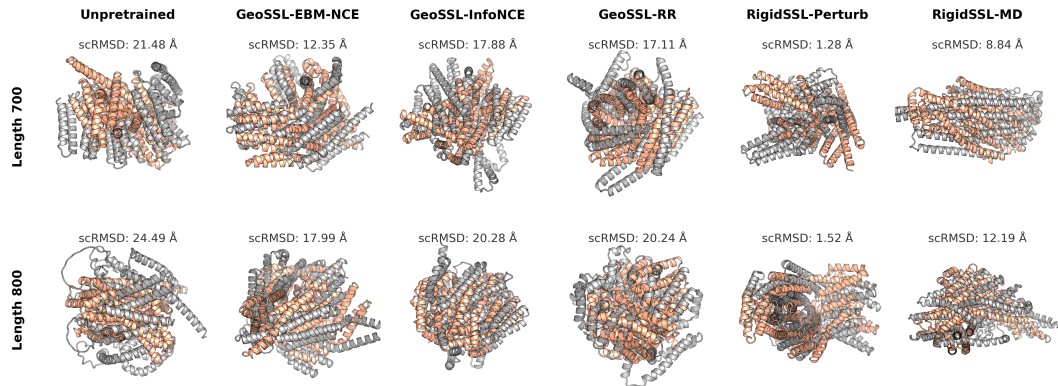

Figure 3: FoldFlow-2 generated structures (orange) compared against ProteinMPNN → ESMFold refolded structures (grey). Columns denote pretraining methods, and rows denote sequence lengths of 700 and 800.

centering coordinates at the motif center of mass. At each timestep, the model predicts vector fields for all residues, but only scaffold coordinates are updated while motif coordinates remain unchanged, thereby sampling scaffolds from a distribution conditioned on the fixed motif geometry.

For each generated backbone scaffold, we perform inverse folding with ProteinMPNN (Dauparas et al., 2022) to design amino acid sequences. We sample $K=8$ sequences per backbone at sampling temperature 0.1. Motif residue identities are fixed during sequence design, and ProteinMPNN designs only the scaffold region. We assess designability by refolding the ProteinMPNN-designed sequences with ESMFold (Lin et al., 2023) and computing structural similarity metrics. A scaffold is designated as designable if at least one of the eight refolded structures satisfies the following success criteria: global C$\alpha$ RMSD < 4.0 Å, motif C$\alpha$ RMSD < 3.0 Å, and scaffold C$\alpha$ RMSD < 4.0 Å. We emphasize that these scRMSD thresholds are relaxed to a reasonable extent relative to the original benchmark to provide greater resolution among methods in the challenging zero-shot setting (Korbeld et al., 2025). Accordingly, results are intended for internal comparison across pretraining frameworks rather than for direct comparison to prior benchmarks in the literature. RMSDs are computed under optimal superposition for each residue subset. We report designability as the fraction of the 100 generated scaffolds per target that satisfy these criteria; results are shown in Table 5.

Table 5: Per-target designability comparison across different geometric pretraining methods. Each entry reports the number of successful designs out of 100 trials for the specified target. Best values are shown in **bold** and second-best are underlined.

| Target | Unpretrained | GeoSSL-EBM-NCE | GeoSSL-InfoNCE | GeoSSL-RR | RigidSSL-Perturb | RigidSSL-MD |
|---|---|---|---|---|---|---|
| 1BCF | 11 | 2 | 29 | 0 | **47** | 15 |
| 1PRW | 5 | 1 | **11** | 1 | **11** | 7 |
| 1QJG | 11 | 9 | **17** | 7 | 8 | 10 |
| 1YCR | 9 | 10 | 8 | 5 | **19** | 12 |
| 2KL8 | 15 | 5 | 23 | 14 | 17 | **34** |
| 3IXT | **25** | 22 | 23 | 8 | **25** | 19 |
| 4JHW | 0 | 0 | **1** | 0 | 0 | 0 |
| 5IUS | 16 | 12 | **36** | 4 | 19 | 22 |
| 5TPN | 0 | 1 | 1 | 0 | **2** | 0 |
| 5TRV_long | 20 | 11 | 30 | 17 | **51** | 25 |
| 5TRV_med | 26 | 17 | **36** | 9 | 22 | 20 |
| 5TRV_short | 16 | **25** | 16 | 8 | 11 | 11 |
| 5WN9 | **12** | 10 | **12** | 0 | 6 | 6 |
| 5YUI | 0 | 0 | 0 | 0 | 0 | **1** |
| 6E6R_long | 9 | 10 | 14 | 8 | **20** | 15 |
| 6E6R_med | 9 | 10 | 6 | 5 | **21** | 7 |
| 6E6R_short | **11** | 5 | 3 | 1 | **11** | 3 |
| 6EXZ_long | 9 | 2 | 7 | 3 | **26** | 5 |
| 6EXZ_med | 4 | 2 | 10 | 2 | **17** | 4 |
| 6EXZ_short | 6 | 2 | **7** | 2 | 5 | **7** |
| 6X93 | 0 | 0 | 0 | **1** | 0 | 0 |
| 7MRX_128 | 0 | 0 | 2 | 2 | **7** | 3 |
| 7MRX_60 | 4 | 2 | 6 | 0 | **12** | 4 |
| 7MRX_85 | 3 | 2 | 2 | 1 | **9** | 2 |
| il7ra_gc | 4 | 3 | **24** | 3 | 3 | 20 |
| rsv_site4 | 18 | 21 | 19 | 5 | **26** | 10 |
| Average | 9.346 | 7.077 | 13.192 | 4.077 | **15.192** | 10.077 |

### F.7 CASE STUDY: GPCR CONFORMATIONAL ENSEMBLE GENERATION

#### F.7.1 TRAINING AND INFERENCE FOR GPCR ENSEMBLE GENERATION

For the case study on GPCR ensemble generation, we utilize the GPCRmd dataset hosted on MDRepo (Rodríguez-Espigares et al., 2020; Roy et al., 2025), which provides all-atom MD simulations across all structurally characterized GPCR classes. To focus on intrinsic receptor flexibility, we specifically use the apo form of each receptor, which was generated by removing the ligand from its binding pocket prior to simulation. Each protein was embedded in a lipid bilayer, solvated with water, and neutralized with ions. All systems were simulated for 500 ns in three replicas using the ACEMD software (Harvey et al., 2009). To generate the training and evaluation samples for RigidSSL, we processed these trajectories into a final dataset comprising 523 training, 68 validation, and 67 testing MD trajectories, each subsampled to 100 frames. We trained AlphaFlow from scratch using the same optimization hyperparameters as the original work, warm-starting the IPA module with pretrained weights. Sampling follows the procedure described in the original work.

#### F.7.2 EVALUATION METRICS FOR GPCR ENSEMBLE GENERATION

To evaluate the generated GPCR ensembles against the ground-truth ensembles, we adopt the metrics used in Jing et al. (2024); we provide brief definitions below and refer the reader to the original work for full details.

Let $\mathcal{X} = \{x^{(1)}, \ldots, x^{(M)}\}$ denote a predicted ensemble of $M$ structures and $\mathcal{Y} = \{y^{(1)}, \ldots, y^{(K)}\}$ the ground-truth MD ensemble of $K$ structures, where each $x^{(m)}, y^{(k)} \in \mathbb{R}^{N \times 3}$ are the 3D coordinates of a protein with $N$ atoms. All structures are RMSD-aligned to a common reference structure prior to analysis.

**Pairwise RMSD.** Mean pairwise $C_\alpha$-RMSD within an ensemble: $\text{PairRMSD}(\mathcal{X}) = \binom{M}{2}^{-1} \sum_{i<j} \text{RMSD}(x^{(i)}, x^{(j)})$.

**All-atom RMSF.** For atom $n$ with ensemble mean $\bar{x}_n$: $\text{RMSF}_n(\mathcal{X}) = \sqrt{M^{-1} \sum_m \|x_n^{(m)} - \bar{x}_n\|^2}$.

**Root Mean $\mathcal{W}_2$-Distance (RMWD).** Fit 3D Gaussians $\mathcal{N}(\mu_n, \Sigma_n)$ to each atom's positional distribution. The squared 2-Wasserstein distance between Gaussians is $W_2^2(\mathcal{N}_1, \mathcal{N}_2) = \|\mu_1 - \mu_2\|^2 + \text{Tr}(\Sigma_1 + \Sigma_2 - 2(\Sigma_1 \Sigma_2)^{1/2})$. Then $\text{RMWD}(\mathcal{X}, \mathcal{Y}) = \sqrt{N^{-1} \sum_n W_2^2(\mathcal{N}[\mathcal{X}_n], \mathcal{N}[\mathcal{Y}_n])}$. This decomposes as $\text{RMWD}^2 = T^2 + V^2$ where $T^2 = N^{-1} \sum_n \|\mu_n^{\mathcal{X}} - \mu_n^{\mathcal{Y}}\|^2$ (translation) and $V^2 = N^{-1} \sum_n \text{Tr}(\Sigma_n^{\mathcal{X}} + \Sigma_n^{\mathcal{Y}} - 2(\Sigma_n^{\mathcal{X}} \Sigma_n^{\mathcal{Y}})^{1/2})$ (variance).

**PCA $\mathcal{W}_2$-Distance.** Project flattened $C_\alpha$ coordinates onto the top 2 principal components and compute $W_2$ (in Å RMSD units) between the projected distributions. *MD PCA*: PCs computed from $\mathcal{Y}$ alone. *Joint PCA*: PCs computed from the equally weighted pooling of $\mathcal{X}$ and $\mathcal{Y}$.

**% PC-sim $> 0.5$.** Let $v_1^{\mathcal{X}}, v_1^{\mathcal{Y}}$ be the top principal components of each ensemble. PC-sim $= |\langle v_1^{\mathcal{X}}, v_1^{\mathcal{Y}} \rangle| / (\|v_1^{\mathcal{X}}\| \|v_1^{\mathcal{Y}}\|)$. We report the percentage of targets with PC-sim $> 0.5$.

**Weak Contacts.** A $C_\alpha$ pair $(i, j)$ is in contact if $\|x_i - x_j\| < 8\,\text{Å}$. Weak contacts are those in contact in the crystal structure but dissociating in $> 10\%$ of ensemble members: $\mathcal{W}(\mathcal{X}) = \{(i,j) : c_{ij}^{\text{ref}} = 1, f_{ij}(\mathcal{X}) < 0.9\}$. Agreement is measured by Jaccard similarity $J(A, B) = |A \cap B|/|A \cup B|$.

**Exposed Residue.** Sidechain SASA is computed via Shrake–Rupley (probe radius $2.8\,\text{Å}$). Cryptically exposed residues are buried in the crystal ($\text{SASA}_n^{\text{ref}} \leq 2.0\,\text{Å}^2$) but exposed ($\text{SASA}_n > 2.0\,\text{Å}^2$) in $> 10\%$ of ensemble members. Agreement is measured by Jaccard similarity.

**Exposed MI Matrix.** For binary exposure indicators $e_n^{(m)} = \mathbf{1}[\text{SASA}_n^{(m)} > 2.0\,\text{Å}^2]$, compute pairwise mutual information $\text{MI}_{ij} = \sum_{a,b} p_{ij}(a, b) \log \frac{p_{ij}(a,b)}{p_i(a) p_j(b)}$. Agreement between MI matrices of $\mathcal{X}$ and $\mathcal{Y}$ is measured by Spearman correlation $\rho$.

Table 6: Evaluation of GPCR MD ensembles generated with pretrained AlphaFlow variants. Best results are shown in **bold**.

| | Metric | Unpretrained | GeoSSL-EBM-NCE | GeoSSL-InfoNCE | GeoSSL-RR | RigidSSL-Perturb (ours) | RigidSSL-MD (ours) |
|---|---|---|---|---|---|---|---|
| Predicting | Pairwise RMSD (=1.55) | 2.37 | 2.32 | 2.31 | 2.48 | **2.20** | 2.66 |
| flexibility | All-atom RMSF (=1.0) | 1.30 | 1.19 | 1.22 | 1.17 | **1.08** | 1.28 |
| | Root mean $\mathcal{W}_2$-dist. ↓ | 18.36 | 18.46 | **17.79** | 17.83 | 17.90 | 18.10 |
| | ↪ Translation contrib. ↓ | 18.26 | 18.31 | **17.75** | 17.76 | 17.83 | 18.05 |
| Distributional | ↪ Variance contrib. ↓ | 1.45 | 1.50 | **1.42** | 1.52 | 1.44 | 1.60 |
| accuracy | MD PCA $\mathcal{W}_2$-dist. ↓ | 3.15 | **2.97** | 3.16 | 3.14 | 3.10 | 3.22 |
| | Joint PCA $\mathcal{W}_2$-dist. ↓ | 18.03 | 17.90 | 17.55 | 17.67 | **17.53** | 17.84 |
| | % PC-sim > 0.5 ↑ | 0.00 | 1.49 | 4.48 | 4.48 | **5.97** | 5.97 |
| Ensemble | Weak contacts $J$ ↑ | 0.33 | 0.35 | 0.36 | 0.36 | 0.33 | **0.43** |
| observables | Exposed residue $J$ ↑ | 0.67 | **0.71** | 0.58 | 0.58 | 0.42 | **0.71** |
| | Exposed MI matrix $\rho$ ↑ | 0.01 | -0.00 | -0.00 | 0.02 | -0.00 | **0.03** |

# G ABLATION STUDIES

We present ablation studies of RigidSSL to isolate the contributions of our pretraining framework from the effects of increased data volume, and to evaluate specific design choices in the view construction process.

First, to disentangle the benefits of our pretraining method from the advantage of simply training on a vastly larger dataset, we compare a FoldFlow-2 baseline trained from scratch on a combined AFDB and PDB dataset against our standard pipeline, which pretrains on AFDB via RigidSSL-Perturb and then finetunes on PDB. As shown in Table 7, RigidSSL achieves higher designability and diversity, along with comparable novelty, while requiring 100,000 fewer training steps. This demonstrates that the observed performance gains stem fundamentally from RigidSSL rather than just an increase in data scale.

Table 7: Comparison of Designability (fraction of proteins with scRMSD ≤ 2.0 Å), Novelty (max. TM-score to PDB), and Diversity (avg. pairwise TM-score and MaxCluster diversity) of structures generated by FoldFlow-2 trained on PDB + AFDB from scratch versus models first pretrained on AFDB and finetuned on PDB. Reported with standard errors. Results for the former are evaluated using the official model checkpoint provided by the authors of Huguet et al. (2024).

| | | | | Designability | Novelty | Diversity | |
|---|---|---|---|---|---|---|---|
| Model | Downstream Dataset | Pretraining | Steps | Fraction (↑) | avg. max TM (↓) | pairwise TM (↓) | MaxCluster (↑) |
| FoldFlow-2 | PDB + AFDB | None | 500k | $0.738 \pm 0.014$ | $0.764 \pm 0.003$ | $0.657 \pm 0.001$ | 0.250 |
| FoldFlow-2 | PDB | RigidSSL-Perturb | 400k | $0.758 \pm 0.016$ | $0.770 \pm 0.003$ | $0.650 \pm 0.001$ | 0.252 |

Second, we investigate the impact of translational and rotational noise scales in RigidSSL-Perturb (Section 3.2.1). As shown in Figure 4, increasing either noise scale leads to more steric clashes and reduced bond validity. The resulting FoldFlow-2 models trained with these noise settings are reported in Table 8.

Table 8: Comparison of Designability (fraction of proteins with scRMSD ≤ 2.0 Å), Novelty (max. TM-score to PDB), and Diversity (avg. pairwise TM-score) of structures generated by FoldFlow-2 pretrained with RigidSSL-Perturb under different noise scales. All metrics include standard errors.

| | | | Designability | Novelty | Diversity |
|---|---|---|---|---|---|
| Translation Noise | Rotation Noise | Length | Fraction (↑) | avg. max TM (↓) | pairwise TM (↓) |
| 0.01 | 0.5 | 100–600 | $0.336 \pm 0.012$ | $0.768 \pm 0.010$ | $0.635 \pm 0.002$ |
| 0.03 | 0.5 | 100–600 | $0.758 \pm 0.016$ | $0.770 \pm 0.003$ | $0.650 \pm 0.001$ |
| 0.05 | 0.5 | 100–600 | $0.589 \pm 0.014$ | $0.769 \pm 0.005$ | $0.654 \pm 0.002$ |
| 0.5 | 0.5 | 100–600 | $0.382 \pm 0.015$ | $0.748 \pm 0.005$ | $0.649 \pm 0.001$ |
| 0.5 | 0.75 | 100–600 | $0.660 \pm 0.014$ | $0.763 \pm 0.004$ | $0.644 \pm 0.001$ |
| 0.5 | 1.0 | 100–600 | $0.352 \pm 0.025$ | $0.772 \pm 0.008$ | $0.614 \pm 0.003$ |
| 1.0 | 0.75 | 100–600 | $0.460 \pm 0.016$ | $0.773 \pm 0.005$ | $0.663 \pm 0.001$ |
| 2.0 | 0.75 | 100–600 | $0.347 \pm 0.014$ | $0.797 \pm 0.006$ | $0.624 \pm 0.002$ |

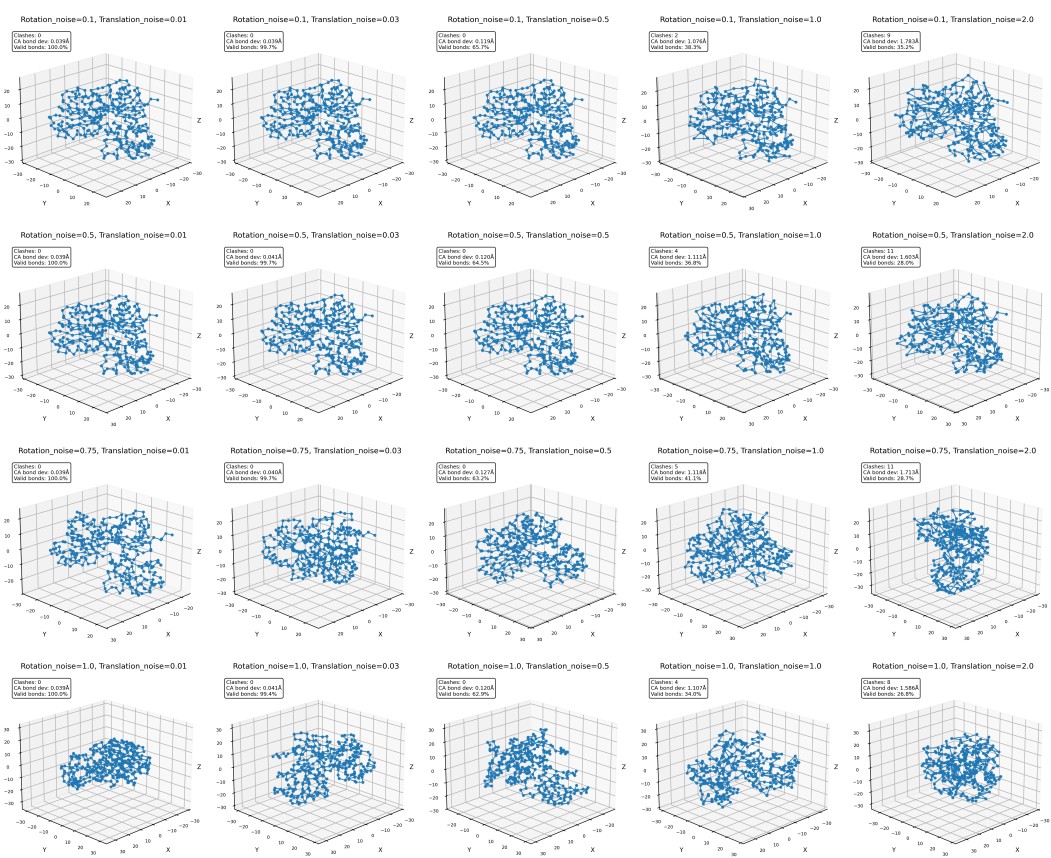

Figure 4: Impact of translation and rotation noise scale on protein structure validity.

