# OpenReview forum: "Rigidity-Aware Geometric Pretraining for Protein Design and Conformational Ensembles"
_ICLR.cc/2026/Conference — ICLR 2026 Poster_

### Official Review · Reviewer_1qeX · 2025-10-28

**Soundness:** 4
**Presentation:** 3
**Contribution:** 3
**Rating:** 6
**Confidence:** 4

**Summary:**

This paper introduces a pretraining method for an unconditional protein backbone structure generation model. Experimental results demonstrate that this approach enhances the performance of FrameDiff and FoldFlow2 in generating long proteins and high-quality proteins that are not purely helical.

**Strengths:**

1. The problem addressed by this method is highly important. A major issue with unconditional protein generation models is that the generated designable proteins are predominantly helical. This method alleviates this problem to some extent through pretraining.

2. The pretraining strategy used in this method is relatively general and trains fast.

3. The experimental section follows established conventions and provides strong support for the claims made in the paper.

**Weaknesses:**

1. Some descriptions are somewhat confusing. For example, it is unclear whether Phase 1 and Phase 2 represent two different strategies or if they are sequentially related.

2. The case study lacks illustrations or examples of designable proteins with a low proportion of helices.

**Questions:**

1. The batch size during the pretraining stage is set to 1, and no analysis related to batch size is provided. Most existing generation models demonstrate that a larger batch size often leads to better training performance. Why a larger batch size was not used in this work?
2. Can this approach naturally extend to the transformer decoder framework like AlphaFold 3?

---

> ### Author Response · Authors · 2025-11-27
> **To Reviewer 1qeX**
>
> Thank you for the thorough and thoughtful review of our submission. We are grateful for your detailed feedback and the time you invested in evaluating our work. Below, we address each of your questions and concerns in turn.
>
> **W1**: Phase 1 and Phase 2 correspond to the two sequential components of the pretraining framework, RigidSSL-Perturb and RigidSSL-MD. We have made this sequencing explicit in the revised manuscript.
>
>
> **W2**: We appreciate the reviewer’s suggestion. We will include additional examples of designable proteins with different secondary-structure compositions in the appendix. We note, however, that the 700- and 800-residue proteins generated by most models tend to exhibit predominantly alpha-helical structures.
>
> **Q1**: We agree that larger batch sizes generally lead to more stable optimization in generative models. However, due to the substantial memory requirements of per-residue SE(3) representations and Invariant Point Attention, which involve storing translations, rotations, and pairwise geometric features for hundreds of residues, our model is inherently memory intensive. Given our initial computational constraints (A100, 40 GB) and to ensure consistency across all experiments, we were able to accommodate only a single full protein structure per batch, corresponding to a batch size of 1. We plan to include additional ablation studies with larger batch sizes once more GPU resources become available.
>
> **Q2**: Yes. While the rigid interpolation strategy is tailored for frame-based protein representations, the underlying multi-view pretraining framework is architecture-agnostic and in principle can be adapted to pretrain other classes of protein generative models. For example, when extending the framework to architectures such as non-equivariant AlphaFold 3, the current rigid interpolation between local residue frames could be replaced with coordinate space interpolation. In this case, interpolation could be performed directly on coordinate distributions, rather than frame-based LERP and SLERP.

---

### Official Review · Reviewer_8zoL · 2025-11-01

**Soundness:** 3
**Presentation:** 2
**Contribution:** 2
**Rating:** 2
**Confidence:** 3

**Summary:**

This paper proposes a rigidity-based geometric pretraining method for protein backbone generation. For perturbation-based view construction on the large-scale AFDB dataset and molecular dynamics trajectories of the ATLAS dataset, they canonicalize proteins into a reference frame and use bi-directional flow matching to pretrain the IPA module, which is shared by FrameDiff, FoldFlow-2.

**Strengths:**

The SE(3) rigidity pretraining for protein backbone generation is reasonable. Perturbations on rigidity align with the protein's natural conformational fluctuations, which can be interpreted as a masking-like paradigm. The MD snapshots used for pretraining are novel and interesting.

**Weaknesses:**

**W1. It is hard to determine whether the performance gains stem from the introduction of new data (e.g., AFDB, ATLAS) or the proposed rigidity-based geometric pretraining method. (My main concern)**

The impact of RigidSSL-MD on diversity has been analyzed in lines 408–411 and Section 5, I think the improvements of diversity are attributed to the new data in the ATLAS dataset.

For RigidSSL-Perturb, both FrameDiff and FoldFlow2 achieved improvements in designability and novelty. However,

(1). FrameDiff was trained only on a small PDB dataset (\~20K), while RigidSSL-Perturb incorporates new data from the AFDB (\~432k).

(2). The training length range for FoldFlow2 is 60-384, while RigidSSL-Perturb ranges is 60-512. For evaluations, the generated length is 100-600. This is unfair.

I suggest authors:

(1). Show FoldFlow-2 + RigidSSL-Perturb results for 100-300 instead of 100-600.

(2). Add a comparison: train baselines Framediff and FoldFlow-2 on the same AFDB and ATLAS datasets using their original training objectives. In this case, comparison on 100-600 is OK.

(3). Use a table in the main body of the paper to present data sources, numbers, protein length ranges, training objectives, training time, and other relevant information of various methods (including training, pretraining, and fine-tuning methods). This will make the paper's contributions clearer.

**Note**: In my opinion, if my understanding is correct, previous CV/NLP tasks usually have scarce data labels, so we adopt self-supervised learning to utilize large amounts of unlabeled data. However, the unconditional protein backbone generation task here is different. We can directly utilize data from AFDB for training the original SE(3) flow matching. FoldFlow-2 [1] also validated that directly using new AFDB data (+PDB, \~160K filtered data in total) for training yields performance improvements. This is why I have the ‘W1’ concern. However, I still believe that the rigidity-based geometric pretraining could be beneficial. Directly learning noise to protein means simultaneously learning the fundamental rules of protein geometry and the complex task of de novo protein design, which may be challenging. RigidSSL decouples these challenges by first pretraining on large-scale data to learn a robust geometric representation that serves as a good initialization. This motivation is reasonable for me, but the authors may need to clarify the specific benefits of RigidSSL according to suggestions (1)-(3).

[1].Sequence-Augmented SE(3)-Flow Matching For Conditional Protein Backbone Generation

**W2. Referring to RigidSSL-Perturb as phase 1 and RigidSSL-MD as phase 2 is confusing. Particularly, they are two independent pre-training methods in the experiment part. What are the results of their combined use? Can you provide further experimental results and analysis?**

**W3. Can you show some results on the conditional generation experiments, e.g.,  motif scaffolding?**


I will raise my score if my concerns are resolved.

**Questions:**

**Q1. I'm curious whether the parameters of these equivariant models based on IPA is hard to scale? For rigidity-based geometric pretraining, thanks to dataset like AFDB, we can scale up the data size during the pretrain phase easily. But can we scale the equivariant models based on IPA module like Proteina[1], SimpleFold[2]?**


[1].Proteina: Scaling Flow-based Protein Structure Generative Models

[2].SimpleFold: Folding Proteins is Simpler than You Think

---

> ### Author Response · Authors · 2025-11-27
> **To reviewer 8zoL (1/2)**
>
> Thank you very much for your thorough and insightful review of our submission. We appreciate the depth of your feedback and the time you spent engaging with our work. Below, we address each of your questions and concerns in turn.
>
> **W1**: To determine whether performance gains stemmed from additional non-PDB data or the proposed pretraining method, we compare FoldFlow-2 trained directly on AFDB+PDB with FoldFlow-2 pretrained on AFDB and then finetuned on PDB, as shown in the table below. Notably, pretraining with RigidSSL-Perturb already achieves higher designability and diversity with 100k fewer steps.
>
>
> |       | Downstream Dataset      | Pretraining        | Steps | Designability (Frac. < 2Å) | Novelty (avg. max TM) | Diversity (pairwise TM) | Diversity (MaxCluster) |
> |-------------|--------------|---------------------|-------|---------------------------|------------------------|---------------------------|--------------------------|
> | FoldFlow2   | PDB + AFDB  | None                | 500k  | 0.738 ± 0.014             | 0.764 ± 0.003          | 0.657 ± 0.001             | 0.250            |
> | FoldFlow2   | PDB   | RigidSSL-Perturb    | 400k  | 0.758 ± 0.016             | 0.770 ± 0.003          | 0.650 ± 0.001             | 0.252           |
>
>
> Additionally, we included pretraining baselines in Table 1 to demonstrate the effectiveness of RigidSSL-Perturb. Specifically, for sample diversity, GeoSSL-InfoNCE was trained on the identical AFDB and ATLAS dataset yet failed to exhibit the diversity improvements observed with RigidSSL. This discrepancy indicates that access to MD data alone is insufficient and a specialized pretraining objective is required to effectively leverage this information, a task for which our rigidity-based algorithm proves particularly effective.
>
>
>
>
> We present the requested comparisons for lengths 100-300 in the table below. Also, we have added a table (Table 1) detailing training and sampling details as Table 3 in our paper.
> |             | Designability (Frac. < 2Å)   | Novelty (avg. max TM)      | Diversity (pairwise TM)   | Diversity (MaxCluster) |
> |------------------|-------------------|------------------|------------------|-----------------------|
> | Unpretrained     | 0.676 ± 0.027     | 0.825 ± 0.004    | 0.605 ± 0.001    | 0.292                 |
> | GeoSSL-EBM-NCE   | 0.800 ± 0.024     | 0.807 ± 0.004    | 0.610 ± 0.001    | 0.248                 |
> | GeoSSL-InfoNCE   | 0.700 ± 0.025     | 0.793 ± 0.005    | 0.627 ± 0.001    | 0.278                 |
> | GeoSSL-RR        | 0.704 ± 0.024     | 0.800 ± 0.005    | 0.597 ± 0.001    | 0.275                 |
> | RigidSSL-Perturb (ours) | 0.944 ± 0.014     | 0.790 ± 0.004    | 0.596 ± 0.002    | 0.309                 |
> | RigidSSL-MD (ours)      | 0.836 ± 0.022     | 0.812 ± 0.004    | 0.595 ± 0.001    | 0.316                 |

---

> ### Author Response · Authors · 2025-11-27
> **To reviewer 8zoL (2/2)**
>
> **W2**: Phase 1 refers to RigidSSL-Perturb, and Phase 2 refers to RigidSSL-MD. These are applied sequentially; Phase 1 is completed before Phase 2 begins. We will adjust our writing to make this explicit.
>
> **W3**: Following RFDiffusion[1], we conducted a motif-scaffolding experiment using FoldFlow2, with the number of designables reported in the table below (out of 100 samples). On average, RigidSSL-Perturb achieves the best performance.
>
> | Target       | Unpretrained | GeoSSL-EBM-NCE | GeoSSL-InfoNCE | GeoSSL-RR | RigidSSL-Perturb (ours) | RigidSSL-MD (ours) |
> |--------------|--------------|----------------|----------------|------------|------------------|-------------|
> | 1BCF         | 11           | 2              | 29             | 0          | 47               | 15          |
> | 1PRW         | 5            | 1              | 11             | 1          | 11               | 7           |
> | 1QJG         | 11           | 9              | 17             | 7          | 8                | 10          |
> | 1YCR         | 9            | 10             | 8              | 5          | 19               | 12          |
> | 2KL8         | 15           | 5              | 23             | 14         | 17               | 34          |
> | 3IXT         | 25           | 22             | 23             | 8          | 25               | 19          |
> | 4JHW         | 0            | 0              | 1              | 0          | 0                | 0           |
> | 5IUS         | 16           | 12             | 36             | 4          | 19               | 22          |
> | 5TPN         | 0            | 1              | 1              | 0          | 2                | 0           |
> | 5TRV_long    | 20           | 11             | 30             | 17         | 51               | 25          |
> | 5TRV_med     | 26           | 17             | 36             | 9          | 22               | 20          |
> | 5TRV_short   | 16           | 25             | 16             | 8          | 11               | 11          |
> | 5WN9         | 12           | 10             | 12             | 0          | 6                | 6           |
> | 5YUI         | 0            | 0              | 0              | 0          | 0                | 1           |
> | 6E6R_long    | 9            | 10             | 14             | 8          | 20               | 15          |
> | 6E6R_med     | 9            | 10             | 6              | 5          | 21               | 7           |
> | 6E6R_short   | 11           | 5              | 3              | 1          | 11               | 3           |
> | 6EXZ_long    | 9            | 2              | 7              | 3          | 26               | 5           |
> | 6EXZ_med     | 4            | 2              | 10             | 2          | 17               | 4           |
> | 6EXZ_short   | 6            | 2              | 7              | 2          | 5                | 7           |
> | 6X93         | 0            | 0              | 0              | 1          | 0                | 0           |
> | 7MRX_128     | 0            | 0              | 2              | 2          | 7                | 3           |
> | 7MRX_60      | 4            | 2              | 6              | 0          | 12               | 4           |
> | 7MRX_85      | 3            | 2              | 2              | 1          | 9                | 2           |
> | il7ra_gc     | 4            | 3              | 24             | 3          | 3                | 20          |
> | rsv_site4    | 18           | 21             | 19             | 5          | 26               | 10          |
> | AVERAGE      | 9.346  | 7.077    | 13.192    | 4.077| 15.192      | 10.077 |
>
> **Q1**: The scaling behavior of IPA is currently not well studied. It is a performative, while expansive module with attention mechanisms that has a quadratic complexity. Due to GPU memory constraints, the current IPA-based generative models strive for a balance between a reasonably large batch size and model size. Notably, linear scaling can be achieved via factorized reformulation, such as FlashIPA[2], which could enable us to scale up IPA-based models. However, this is beyond the scope of the current work, and we would like to explore this question in future work.
>
>
> [1]. De novo design of protein structure and function with RFdiffusion
>
> [2]. Flash Invariant Point Attention

---

### Official Review · Reviewer_yn5Y · 2025-11-02

**Soundness:** 3
**Presentation:** 3
**Contribution:** 2
**Rating:** 2
**Confidence:** 4

**Summary:**

This paper proposes a pretraining method for protein generation by viewing each residue structure as a rigid body and using flow matching. Specifically, it first defines an inertial frame as a reference frame for each protein and aligns protein backbone structure with the inertial frame; then, it perturbs each protein's translation and rotation by randomly sampling transforms from an Euclidean space and a special orthogonal group to form two views; next, it samples trajectory segments from a molecular dynamics dataset; finally, it uses flow matching to generate each view from the other views. The experiments are conducted on a protein generation task and the results outperform the state-of-the-art methods.

**Strengths:**

1. It utilizes the structural information available in large-scale protein datasets to pretrain a protein generation model in an unsupervised manner.

2. It achieves superior protein generation performance to the compared approaches.

**Weaknesses:**

1. This paper is more engineering-oriented. Though it achieves superior performance across several models on a protein generation benchmark, it seems to contain little new algorithms or architectures. Reference frame definition and flow matching are widely used across many areas, including, but not limited to, machine learning, computer vision, and computational biology.

2. The construction of two different conformation views is a little bit new, but the motivation for such a construction is unclear. We can directly sample two time steps in a dynamic trajectory and generate protein structures at one time step to the other.

**Questions:**

1. In Section 3.2.1, $g^0$ and $g^1$ represent the original state of one protein and the perturbed state by adding random noise; in Section 3.2.2, $g^0$ and $g^1$  represent different conformations sampled at different time steps along a dynamic trajectory. This inconsistency is confusing.

2. In Section 3.2.1, the other view is constructed by adding random noises sampled from Euclidean space and the SO(3) group. How to make sure the perturbed view is biologically valid?

---

> ### Author Response · Authors · 2025-11-27
> **To reviewer yn5Y**
>
> Thank you for offering a thoughtful and insightful review of our paper. We value the detailed feedback you provided and the time you devoted to assessing our work. Below, we address each of your questions and concerns in turn.
>
> **Concern 1: this paper is engineering-oriented with little new algorithms or architectures**
>
> While we build on established concepts, our core contribution is the novel synthesis of these components into a rigidity-aware framework specifically designed for geometric pretraining for protein structures. Our novelty lies in: 1. The introduction of a two-phase view construction, combining manifold-aware SE(3) perturbations (RigidSSL-Perturb) with physical dynamics from MD trajectories (RigidSSL-MD), 2. The canonicalization of protein structure to inertial frame, 3) a rigid-body flow matching objective formulated to maximize mutual information between dynamic protein structures.
>
> **Concern 2: Trajectory sampling is a better-motivated alternative to RigidSSL-Perturb's view construction.**
>
> Trajectory sampling is exactly what’s performed in RigidSSL-MD. However, obtaining trajectories is nontrivial as conducting MD simulations for all 542K proteins in AFDB is computationally infeasible for us. RigidSSL-Perturb aims to circumvent this via simulating the fast, functionally neutral atomic vibrations in protein structures, which approximates protein dynamics while preserving structural validity.
>
>
> **Concern 3: notation for g0 and g1**
>
> We thank the reviewer for this query, as it allows us to clarify our notation. The use of g0 and g1 in both sections is intentional. In our framework, g0 and g1 always represent a pair of correlated views for our multi-view pretraining objective. The two sections (3.2.1 and 3.2.2) simply describe the different mechanisms for generating these views (i.e., perturbation vs. MD sampling). We intentionally share the notation to highlight that their role in our pretraining objective function is identical.
>
> **Concern 4: structural validity in RigidSSL-Perturb**
>
> The noise level was calibrated to maintain structural validity (see Figure 4 in Appendix G), yielding perturbed structures that exhibit near minimal bond invalidity and steric clashes.

---

### Official Review · Reviewer_LsQ9 · 2025-11-07

**Soundness:** 2
**Presentation:** 3
**Contribution:** 2
**Rating:** 2
**Confidence:** 4

**Summary:**

This work proposes a pre-training procedure for diffusion and flow-matching based protein design methods. The pre-training method consists of two phases. The first phase, RigidSSL-Perturb, adds Gaussian noise to the translation and rotation component of the residual frames of a protein's backbone. The second phase, RigidSSL-MD, uses pairs of structures from the ATLAS dataset, which contains molecular dynamics simulations of proteins.

**Strengths:**

Incorporating MD simulations into a pre-training step of protein design methods is an interesting and novel contribution.

**Weaknesses:**

The experiments demonstrate that RigidSSL-Perturb outperforms the baselines for designability and novelty, while RigidSSL-MD outperforms the baselines in diversity. However, RigidSSL-MD is not the methods of choice with respect to designability and novelty. These results limit the appilcability of the approach, as there seems no practical advantage of RigidSSL-MD over RigidSSL-Perturb, which is merely a simple data-augmentation of the data with Gaussian noise.

The examples of generated structures in figure 3 for RigidSSL seem to solely consist of alpha helices, a common bias in generative models for proteins (compare e.g. with Wagner et al. 2024). Please also report alpha helix and beta strand content for each method.

References:
Wagner, S., Seute, L., Viliuga, V., Wolf, N., Gräter, F., & Stühmer, J. (2024). Generating highly designable proteins with geometric algebra flow matching. Advances in Neural Information Processing Systems, 37, 77987-78026.

**Questions:**

Is high designability of RigidSSL-Perturb potentially only achieved by a high amount of alpha helices?

The augmentation of RigidSSL-Perturb, which adds Gaussian noise to the coordinates and rotation, could destroy physical plausibility of the proteins. How is the noise level controlled to prevent this?

---

> ### Author Response · Authors · 2025-11-27
> **To reviewer LsQ9**
>
> Thank you for your thorough and insightful review of our submission. We appreciate the detailed feedback and the time you dedicated to engaging with our work. Below, we respond to each of your questions and concerns in turn.
>
> **Concern 1: Applicability of RigidSSL-MD**
>
> RigidSSL-MD leads to the most diverse set of samples, while maintaining a reasonably high level of designability and novelty (eg. second to the best in FoldFlow2). Given the gap between computational oracles and real world wet lab validation, generating a structurally varied set of candidates is especially valuable as it increases the probability of identifying functional hits while avoiding the resource-intensive failure mode of testing redundant, non-viable designs.
>
> **Concern 2: RigidSSL-Perturb as a simple data-augmentation**
> **& Question 2: physical plausibility of perturbed structures**
>
> We would like to clarify that the perturbation is not intended as a data augmentation technique for the downstream protein generative models; rather, it serves as the view-construction mechanism for our pretraining objective. This follows a standard paradigm in SSL, analogous to masking tokens in NLP. The aim is not to introduce noisy data, but to learn representations that are invariant to minor, physically plausible conformational fluctuations, using carefully chosen noise levels that preserve structural validity (see Figure 4 in Appendix G).
>
> **Concern 3: Secondary structure diversity in the generated samples**
>
> We would like to note that the purpose of the case study in Figure 3 is to illustrate each method's best results from a special design case at ultra-long residue lengths (700 and 800), where RigidSSL is the only method that generates designable sequences while all methods display a bias towards alpha helices. As per the reviewer’s request, we report the secondary structure distribution for each method in the case study in the table below.
> |         | Length | α-helix | β-sheet | Coil   | scRMSD (Å) |
> |-------------------|--------|---------|---------|--------|------------|
> | Unpretrained          | 700    | 0.921   | 0.000   | 0.0786 | 21.48      |
> | GeoSSL-EBM-NCE    | 700    | 0.886   | 0.000   | 0.114  | 12.35      |
> | GeoSSL-InfoNCE    | 700    | 0.886   | 0.000   | 0.114  | 17.88      |
> | GeoSSL-RR         | 700    | 0.910   | 0.000   | 0.0900 | 17.11      |
> | RigidSSL-Perturb  | 700    | 0.900   | 0.00143 | 0.0986 | 1.28  |
> | RigidSSL-MD       | 700    | 0.907   | 0.000   | 0.0929 | 8.84       |
> | Unpretrained          | 800    | 0.918   | 0.000   | 0.0825 | 24.49      |
> | GeoSSL-EBM-NCE    | 800    | 0.895   | 0.000   | 0.105  | 17.99      |
> | GeoSSL-InfoNCE    | 800    | 0.899   | 0.000   | 0.101  | 20.28      |
> | GeoSSL-RR         | 800    | 0.916   | 0.000   | 0.0838 | 20.24      |
> | RigidSSL-Perturb (ours)  | 800    | 0.899   | 0.000   | 0.101  | 1.52   |
> | RigidSSL-MD (ours)       | 800    | 0.891   | 0.000   | 0.108  | 12.19      |
>
> For the common design space of 100-600 residues, both RigidSSL-Perturb and RigidSSL-MD generate significantly more diverse secondary structures, including coils and mixed α-helix/β-sheet compositions as shown in the table below.
> |              | α-helix   | β-sheet              | Coil               | RMSD (Å)|
> |-------------------|--------------------|--------------------|--------------------|-------|
> | AF2 (Pretraining) | 0.416 ± 0.186      | 0.188 ± 0.128      | 0.396 ± 0.122      | N/A   |
> | PDB (Training)    | 0.370 ± 0.226      | 0.241 ± 0.170      | 0.389 ± 0.108      | N/A   |
> | Unpretrained          | 0.874 ± 0.029      | 0.000 ± 0.000      | 0.126 ± 0.029      | 0.720 |
> | GeoSSL-EBM-NCE           | 0.878 ± 0.026      | 0.000 ± 0.000      | 0.122 ± 0.026      | 0.732 |
> | GeoSSL-InfoNCE           | 0.860 ± 0.037      | 0.001 ± 0.016      | 0.138 ± 0.033      | 0.784 |
> | GeoSSL-RR                | 0.861 ± 0.074      | 0.007 ± 0.049      | 0.132 ± 0.037      | 0.736 |
> | RigidSSL-Perturb (ours)  | 0.840 ± 0.109      | 0.019 ± 0.077      | 0.141 ± 0.042      | 0.807 |
> | RigidSSL-MD (ours)     | 0.724 ± 0.184      | 0.107 ± 0.144      | 0.169 ± 0.052      | 0.999 |

---

### Author Response · Authors · 2025-12-04
**To Area Chairs, Reviewers, and Readers**

As the discussion period concludes, we thank the reviewers for their constructive feedback. Below is a summary of the key outcomes from the rebuttal phase, highlighting how we have addressed the primary concerns regarding performance and method clarity.

### 1. Positive Highlights and Consensus
We are encouraged that the reviewers recognized the value and novelty of our approach. Reviewers agreed that RigidSSL is a reasonable framework that addresses a highly important problem in protein design (Reviewer 1qeX and 8zoL). The incorporation of MD simulations into pretraining was highlighted as interesting and novel (Reviewers LsQ9 and 8zoL). Reviewers noted that RigidSSL achieves "superior protein generation performance" compared to previous state-of-the-art approaches (Reviewer yn5Y).

### 2. Additional Analysis and Empirical Validation
We provided new experimental data to specifically resolve concerns regarding the source of performance gains and the quality of generated structures.

* **Decoupling Method from Data Scale:** To address whether gains stemmed from the method or simply the data size, we compared our method against the baseline FoldFlow-2 trained on the **exact same dataset** (AFDB + PDB). RigidSSL-Perturb outperforms this data-matched baseline while requiring **100k fewer training steps**. This confirms that the performance gain is strictly methodological.

* **Secondary Structural Diversity:** We clarified concerns regarding "helix bias." For standard lengths (100–600 residues), RigidSSL generates diverse secondary structures, including α-helix, β-sheet, and coil compositions, that match the PDB distribution more closely than baselines. We clarified that the 'helix bias' observed in Figure 3 was specific to the long-chain case study, where RigidSSL-Perturb was the only method capable of generating designable structures.

* **Conditional Generation:** To demonstrate robustness beyond unconditional generation, we conducted an additional evaluation on motif scaffolding across 26 targets. RigidSSL-Perturb achieved the highest success rate compared to baselines, validating the transferability and downstream utility of our pretrained representations.

### 3. Clarity and Correctness
We clarified several aspects of the manuscript to ensure the method is accurately understood:
* **Sequential Phases:** We clarified that Phase 1 (Perturb) and Phase 2 (MD) are applied **sequentially**, not independently.
* **Physical Validity:** We demonstrated that our perturbation noise levels are calibrated to maintain structural validity, resulting in minimal steric clashes.
* **Notation:** We unified the notation across sections to emphasize that both view construction methods optimize a shared pretraining objective.


We believe these revisions and new experiments solidly confirm that RigidSSL offers a robust, methodologically distinct improvement over existing geometric pretraining baselines.

---

### Meta-Review · Area_Chair_a6Br · 2026-01-06

**Summary:**

The reviewers have concerns about novelty, clarity, whether the improvements come from a larger training set or the pretraining phase, and whether the method can generate designable structure with beta sheets. In my opinion, while the initial scores are generally very low, all concerns are addressed very well by the rebuttal.

**Reviewer Concerns:**

**Lack of methodological novelty.**

The rebuttal clarifies that the SSL objective and components are novel. In my opinion, this is not a fair criticism of the paper.

**Mostly generate alpha helices**

This is only true for long generations, and on generation lengths that are represented in training, RigidSSL actually shifts the secondary structure distribution to better match that of natural structures.

**Clarity, especially on phase 1/phase 2 vs Perturb/MD.**

The authors address this by consistently using Perturb/MD.

**RigidSSL-MD seems to hurt designability and novelty.**

The rebuttal clarifies that RigidSSL-MD improves diversity. In addition, I would like to add that it's interesting that large-scale random perturbation is more effective here than using MD snapshots -- having more interesting results is not a weakness.


**It is hard to determine whether the performance gains stem from the introduction of new data (e.g., AFDB, ATLAS) or the proposed rigidity-based geometric pretraining method.**

The rebuttal shows that RigidSSL outperforms just training on AFDB + PDB.

**Reviewer Scores:**

LsQ9: 2 -> 8. Has concerns about secondary structure content and the performance of RigidSSL-MD, which are more than sufficiently addressed.

yn5Y: 2 -> 8. Concerns are about novelty, which in my opinion are unfounded.

8zoL: 2 -> 6. Main concern was about disentangling the effects of data scale and pretraining task, which was addressed. Has additional requests for additional tasks (motif scaffolding) and scaling to more recent models such as la-proteina.

1qeX: 6 -> 8. Has concerns about clarity and being secondary structure content, which are sufficiently addressed.

---

### Decision · Program_Chairs · 2026-01-26

Accept (Poster)